# Preserving Fairness in AI under Domain Shift

## Abstract

Existing algorithms for ensuring fairness in AI use a single-shot training strategy, where an AI model is trained on an annotated training dataset with sensitive attributes and then fielded for utilization. This training strategy is effective in problems with stationary distributions, where both the training and testing data are drawn from the same distribution. However, it is vulnerable with respect to distributional shifts in the input space that may occur after the initial training phase. As a result, the time-dependent nature of data can introduce biases and performance degradation into the model predictions. Model retraining from scratch using a new annotated dataset is a naive solution that is expensive and time-consuming. We develop an algorithm to adapt a fair model to remain fair and generalizable under domain shift using solely new unannotated data points. We recast this learning setting as an unsupervised domain adaptation (UDA) problem. Our algorithm is based on updating the model such that the internal representation of data remains unbiased despite distributional shifts in the input space. We provide empirical validation on three common fairness datasets to show that the challenge exists in practical setting and to demonstrate the effectiveness of our algorithm.

## 1   Introduction

AI has been extensively utilized in automating heavy and electric industry tasks such as logistics, transportation, retail, e-commerce, entertainment and gaming. This growing reliance on AI, particularly deep learning, owes much to its ability to handle vast datasets and bypass tedious feature engineering. This success has spurred the application of deep learning approaches in critical decision-making areas such as loan approvals, parole verdicts, healthcare, and police assignments Chouldechova & Roth (2018). However, these methods often focus on maximizing some performance metric and compared to fundamental statistical approaches, lack explainability in their decisions. Based on inherent biases present in the data, this can translate into features such as race, sex or age influencing outcomes.

It is well documented that some of the best AI models are biased against certain racial or gender sub-groups Eidinger et al. (2014); Zhang et al. (2017); Cirillo et al. (2020) and can produce adverse outcomes for disadvantaged groups. Hence, fairness is a major concern for using AI in societal decision-making processes. This concern is particularly important in deep learning because data-driven learning can unintentionally lead to training unfair models due to the inherent biases that exist in annotating training datasets by human workers or skewed data distributions conditioned on certain sensitive attributes Buolamwini & Gebru (2018). As a result, training models by simply minimizing the empirical error on relevant datasets may add spurious correlations between majority subgroup features and positive outcomes for them. This unwanted outcome happens because statistical learning primarily discovers correlations rather than causation. Thus, the decision boundary of AI models may be informed by group-specific characteristics that are irrelevant to the decision task Dua & Graff (2017). For example, since the income level is generally correlated positively with the male gender, it can lead to training models with unfair decisions against female loan applicants.

The crucial concern about fairness in AI and the need to overcome the resulting adverse effects have resulted in significant research interest from the AI community. The first attempt to address bias in AI is to arrive at a commonly agreed-upon definition of fairness. Pioneer works in this area focused on defining quantitative notions for fairness based on commonsense intuition and using them to quantitatively demonstrate the presence and severity of bias in AI Buolamwini & Gebru (2018); Caliskan et al. (2017). Most existing fairness metrics consider that the input data points possess characteristics of protected subgroups Feldman et al. (2015), e.g., gender and race, in addition to standard features that are used for model training based on empirical risk minimization (ERM). Based on subgroup membership, majority and minority populations emerge, or in general subgroups, which can be used to define fairness

metrics. A model is then assumed to be fair if its predictions possess a notion of probabilistic independence for data membership into the subgroups Mehrabi et al. (2021) (see Section 5.1.3 for definitions of common fairness metrics).

Fairness in an AI models can be reinforced by mapping data into a latent space in which data representations are independent from the sensitive attributes. For example, we can benefit from adversarial learning for this purpose Zhang et al. (2018). Since the sensitive attributes are absent in the latent space, decision-making will not consider sensitive attributes. Despite being an effective approach, most existing fair model training algorithms consider that the data distribution will remain stationary after the training stage. This assumption is rarely true in practical settings, particularly when a model is used over extended periods, because societal applications are dynamic but fairness metrics are normally static. As a result, a fair model might fail to maintain its fairness under the input-space distributional shifts or when the model is used on differently sourced tasks Pooch et al. (2019). The naive solution of retraining the model after distributional shifts requires annotating new data points to build datasets representative of the new input distribution. This process, however, is time consuming and expensive for deep learning and is challenging when data annotation becomes a persistent task. As a result, it is highly desirable to develop algorithms that can preserve model fairness under distribution shifts. Unfortunately, this problem has been marginally explored in the AI literature.

The negative effect of distributional shifts on the performance of AI models is well-known and the problem of model adaptation has been investigated extensively in the unsupervised domain adaptation (UDA) literature Tzeng et al. (2017); Daumé III (2009). The goal in UDA is to train a model with a good generalization performance on a target domain, where only unannotated data is available. The idea is to transfer knowledge from a related source domain, where annotated data is accessible. A primary group of UDA algorithms achieves this goal by matching the source and the target distributions in a shared embedding space Redko et al. (2017) such that the embedding space is domain-agnostic. As a result, a classifier that receives its input from the embedding space will generalize well in the target domain, despite being trained solely using the source domain annotated data. To align the two distributions in such an embedding, data points from both domains are mapped into a shared feature space that is modeled as the output space of a deep neural encoder. The deep encoder is then trained to minimize the distance between the two distributions, measured in terms of a suitable probability distribution metric. However, existing UDA algorithms overlook model fairness and solely consider improving model performance in the target domain. In this work, we adopt the idea of domain alignment in UDA to preserve model fairness and mitigate model biases introduced by domain shift.

**Contribution:** We address the problem of preserving the model fairness and the model generalization under distributional shifts in the input space when only unannotated data is accessible after an initial training stage. We model this problem within the classic unsupervised domain adaptation paradigm. Our specific contributions include:

- We develop an algorithm that minimizes distributional mismatches that results from domain shift in a shared embedding space to maintain model fairness and model performance in non-stationery learning settings.

- We build three AI tasks using three standard fairness benchmarks and demonstrate that in addition to model performance, model fairness is compromised when domain shift exists in real-world applications.

- We conduct extensive empirical explorations and demonstrate that the existing methods for fairness in AI are vulnerable in our learning setting and show that the proposed algorithm is effective.

## 2 Related Work

### 2.1 Fairness in AI

There are various approaches for training a fair model for a single domain. A primary idea in existing works is to map data points into an embedding space at which the sensitive attributes are entirely removed from the representative features, i.e., an attribute-agnostic space. As a result, a classifier that receives its input from this space will make unbiased decisions due to the independence of its decisions from the sensitive attributes. After training the model, fairness can also be measured at the classifier output using a desired fairness metric. Ray et al. 2020 develop a fairness algorithm that induces probabilistic independence between the sensitive attributes and the classifier outputs by minimizing the optimal transport distance between the probability distributions conditioned on the sensitive attributes. Hence, the transformed probability in the embedding space then becomes independent (unconditioned) from the sensitive attributes. Celis et al. 2019b study the possibility of using a meta-algorithm for fairness with respect to several

disjoint sensitive attributes. Du et al. 2021 have followed a different approach. Instead of training an encoder that removes the sensitive attributes in a latent embedding space and then training a classifier, they propose to debias the classifiers by leveraging samples with the same ground-truth label yet having different sensitive attributes. The idea is to discourage undesirable correlation between the sensitive attribute and predictions in an end-to-end scheme, allowing for the emergence of attribute-agnostic representations in the hidden layers of the model. Agarwal et al. 2018 propose an approach that incrementally constructs a fair classifier by solving several cost-constrained classification problems and combining results. Zhang et al. 2018 train a deep model to produce predictions independent of sensitive attributes by training a classifier network to predict binary outcomes and then inputting the predictions to an adversary that attempts to guess their sensitive attribute. By optimizing the network to make this task harder for the adversary, their approach leads to fair predictions. Beutel et al. 2017 benefit from removing sensitive attributes to train fair models by indirectly enforcing decision independence from the sensitive attributes in a latent representation using adversarial learning. They also amend the encoder model with a decoder to form an autoencoder. Since the representations are learned such that they can self-reconstruct the input, they become discriminative for classification purposes as well. These work consider stationary settings. Our work builds upon using adversarial learning to preserve fairness when distribution shifts exist. In order to combat domain shift, our idea is to additionally match the target data distribution with the source data distribution in the latent embedding space, a process that ensures classifier generalization.

## 2.2 Unsupervised Domain Adaptation

Works on domain alignment for UDA follow a diverse set of strategies. The goal of existing works in UDA is solely to improve the prediction accuracy in the target domain in the presence of domain shift without exploring the problem of fairness. The closest line of research to our work addresses domain shift by minimizing a probability discrepancy measure between two distributions in a shared embedding space. Selection of the discrepancy measure is a critical task for these works. Several UDA methods simply match the low-order empirical statistics of the source and the target distributions as a surrogate for the distributions. For example, the Maximum Mean Discrepancy (MMD) metric is defined to match the means of two distributions for UDA Long et al. (2015; 2017). Correlation alignment is another approach to include second-order moments Sun & Saenko (2016). Matching lower-order statistical moments overlooks the existence of discrepancies in higher-order statistical moments. In order to improve upon these methods, a suitable probability distance metric can be incorporated into UDA to consider higher-order statistics for domain alignment. A suitable metric for this purpose is the Wasserstein distance (WD) or the optimal transport metric Courty et al. (2016); Bhushan Damodaran et al. (2018). Since WD possesses non-vanishing gradients for two non-overlapping distributions, it is a more suitable choice for deep learning than more common distribution discrepancy measures, e.g., KL-divergence. Optimal transport can be minimized as an objective using first-order optimization algorithms for deep learning. Using WD has led to a considerable performance boost in UDA Bhushan Damodaran et al. (2018) compared to methods that rely on aligning the lower-order statistical moments Long et al. (2015); Sun & Saenko (2016).

## 2.3 Domain Adaptation in Fairness

Works on benefiting from knowledge transfer to maintain fairness are relatively limited. Madras et al. 2018a benefit from adversarial learning to learn domain-agnostic transferable representations for fair model generalization. Coston et al. 2019 consider a UDA setting where the sensitive attributes for data points are accessible only in one of the source or the target domains. Their idea is to use a weighted average to compute the empirical risk and then tune the corresponding data point-specific weights to minimize co-variate shifts. Schumann et al. 2019 consider a similar setting, where they define the fairness distance of equalized odds, and then use it as a regularization term in addition to empirical risk, minimized for fair cross-domain generalization. Hu et al. 2019 address fairness in a distributed learning setting, where the data exist in various servers with private demographic information. Singh et al. 2021 consider that a causal graph for the source domain data and anticipated shifts are given. They then use feature selection to estimate the fairness metric in the target domain for model adaptation. Zhang and Long 2021 explore the possibility of training fair models in the presence of missing data in a target domain using a source domain with complete data and find theoretical bounds for this purpose. Yoon et al. 2020 consider a fair adaptation scenario where a fair classifier trained on a source domain is deployed on a target domain where the sensitive attribute changes. Oneto et al. 2020 propose to improve model fairness and generalization to new domains by framing the fair transfer learning problem in a multi-task learning framework. Pham et al. 2023 propose a multi source fairness preserving approach, where an algorithm leverages several source domains in order to ensure fairness and generalization on a target domains.

138 Our learning setting is relevant yet different from the above settings. We consider a standard UDA setting where the
139 sensitive attributes are accessible in both domains. The challenge is to adapt the model to preserve fairness in the
140 target domain without requiring data annotation when domain shift occurs.

## 3 Problem Formulation

142 We first describe how to train a fair model, then explain how the problem extends to a non-stationery setting, and
143 offer our solution in the next section. Consider a source domain $\mathcal{S}$, where we are given an annotated training dataset
144 $\mathcal{D}^s = (X^s, A^s, Y^s) \in \mathbb{R}^{N \times d} \times \{0,1\}^N \times \{0,1\}^N$ for which $X^s \in \mathbb{R}^n$ represents feature vectors with dimension
145 $d$ and $Y^s$ represents the binary labels. Additionally, $A^s$ represents binary sensitive attributes for each data point,
146 e.g., race, sex, age, etc. Each triplet $(\boldsymbol{x}^s, \boldsymbol{a}^s, \boldsymbol{y}^s)$ is drawn from the source domain distribution $P_{\mathcal{S}}(\boldsymbol{X}, \boldsymbol{A})$, where
147 the feature vector corresponds to characteristic features that are used for decision-making, e.g., occupation length,
148 education years, credit history, etc. Our goal is to train a fair model with respect to the sensitive attributes, e.g., sex,
149 race, etc. to perform binary decision making, e.g., approving for a loan, parole in prison system, etc.

150 In classic parametric supervised learning, we select a family of predictive functions $f_\theta : (X^s, A^s) \rightarrow Y^s$, parameter-
151 ized with learnable parameters $\theta$. We then search for the model with the optimal parameter based on ERM on the fully
152 annotated dataset $\mathcal{D}^s$, as a surrogate for a model with the expected error on the unknown source domain distribution:

$$\hat{\boldsymbol{\theta}} = \arg \min_{\boldsymbol{\theta}} \mathcal{L}_{sl} = \arg \min_{\boldsymbol{\theta}} \{ \frac{1}{N} \sum_{i=1}^{N} \mathcal{L}_{bce}(f_{\boldsymbol{\theta}}(\boldsymbol{x}^s, \boldsymbol{a}^s), \boldsymbol{y}^s) \}, \tag{1}$$

153 where $\mathcal{L}_{bce}$ is a suitable loss function such a binary cross-entropy loss. Under certain conditions, solving equation 1
154 leads to training a generalizable model during the testing stage. However, there is no guarantee to obtain a fair model
155 because only prediction accuracy is optimized in equation 1. Inherent bias in the training dataset, e.g., over/under-
156 representation of subgroups, can lead to training a biased model. Note that although the sensitive attributes are not
157 used in equation 1, the sensitive attribute may still be highly correlated with the decision features due to data collection
158 procedures. For example, a human operator might have subconsciously consider a sensitive attribute for annotation.

159 An effective approach to train a fair model is to map the domain data into a latent embedding space such that the
160 encoded data representations are fully independent from the sensitive attributes $A$. There are various approaches
161 to implement this idea via training an appropriate encoding function. Inspired by adversarial learning, a group of
162 fairness algorithms rely on solving a min-max optimization problem for this purpose Beutel et al. (2017); Madras
163 et al. (2018b); Zhang et al. (2018). To this end, we first consider that the end-to-end predictive model $f_\theta(\cdot) : \mathbb{R}^d \rightarrow \mathbb{R}^2$
164 can be decomposed into an encoder subnetwork $e_u(\cdot) : \mathbb{R}^d \rightarrow \mathbb{R}^z$, with learnable parameters $u$, followed by a classifier
165 subnetwork $g_v(\cdot) : \mathbb{R}^z \rightarrow \mathbb{R}^2$ with learnable parameters $v$, where $f_\theta(\cdot) = (g_v \circ e_u)(\cdot)$ and $\theta = (u, v)$. The parameter
166 $z$ denotes the dimension of the latent embedding space that we want to be sensitive-agnostic which is modeled as the
167 output space of the encoder subnetwork. To induce "independence from the sensitive attribute" in the latent space, we
168 consider an additional classification network $h_w(\cdot) : \mathbb{R}^z \rightarrow \mathbb{R}^2$ with learnable parameters $w$. This classifier is tasked
169 to predict the corresponding sensitive attribute $\boldsymbol{a}^s$ using the latent space representations $e_u(\boldsymbol{x}^s, \boldsymbol{a}^s)$.

170 The core idea is to induce "probabilistic independence from sensitive attributes" by training $e_u(\cdot)$ and $h_w(\cdot)$ in an ad-
171 versarial learning scheme, where $e_u(\cdot)$ plays the role of the generator network and $h_w(\cdot)$ is the discriminator network.
172 In other words, if the latent representations are independent from the sensitive attribute, $A$, the classifier $h(\cdot)$ would
173 perform poorly. To this end, consider the loss function for predicting the sensitive attributes:

$$\mathcal{L}_{fair}^s = \mathcal{L}_{bce}((h_w \circ e_u)(\boldsymbol{x}^s, \boldsymbol{a}^s), \boldsymbol{a}^s). \tag{2}$$

174 To train an attribute-agnostic encoder, we solve the following alternating min-max optimization process to train a fair
175 model based on adversarial learning scheme Goodfellow et al. (2014):

176    1. We fix the encoder $e_u(\cdot)$ and minimize the fairness loss $\mathcal{L}_{fair}$ through updating the attribute classifier $h_w(\cdot)$.

177    2. We then fix the attribute classifier $h_w(\cdot)$ and maximize the fairness loss $\mathcal{L}_{fair}$ by updating the encoder $e_u(\cdot)$.

178 The first step will perform ERM for the attribute prediction classifier, conditioned on the encoder network being fixed.
179 The second step will keep the classifier fixed and ensures that the latent data representations are as little informative

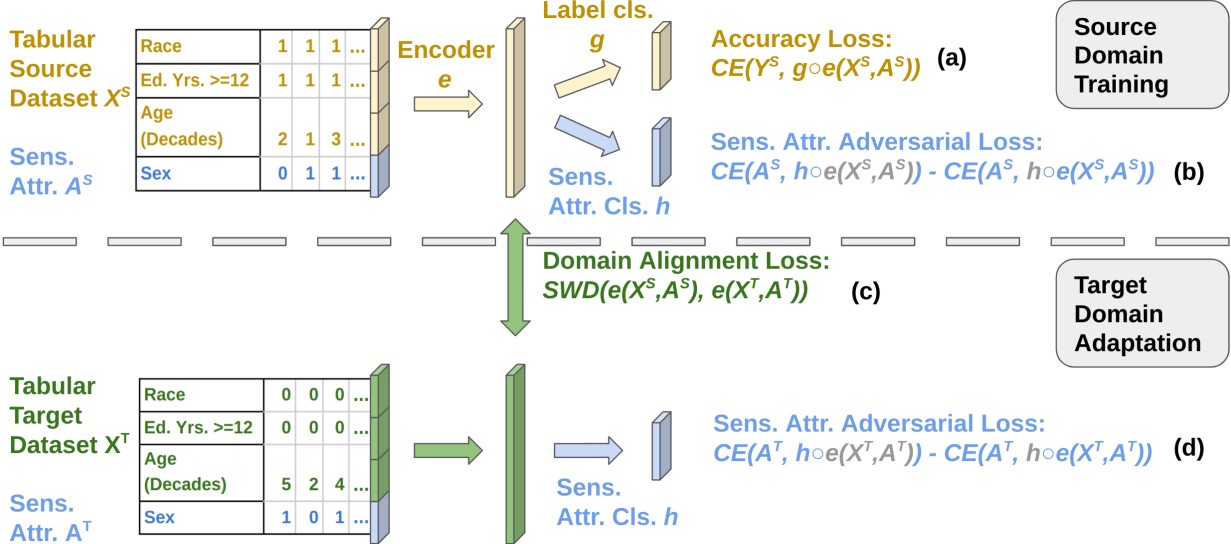

Figure 1: Block-diagram description of the proposed framework for preserving fairness under domain shift. First, a fair model is trained on a fully labeled source domain: (a) minimizing binary cross entropy loss against the source labels (Eq. 1) ensures the learnt embeddings are informative with respect to the classification task (b) adversarial optimization with respect to the sensitive attribute (Eq. 2) makes the learnt embeddings conditionally independent from the sensitive attributes. During adaptation on the unlabeled target domain: (c) Sliced Wasserstein Distance is minimized between the target embedding distribution and the source embedding distribution (Eq. 4) in order to maintain the relevance of the source classifier on the target domain, (d) the fairness loss is also minimized on the target domain to ensure conditional independence of the embeddings and sensitive attributes.

as possible about the sensitive attribute $A$. Similar to vanilla adversarial learning, empirical explorations demonstrate that the above iterative alternations between the two optimization steps will lead to training an encoder that produces latent representations that are independent from the sensitive attribute when the attribute classifier fails to predict the sensitive attributes. To train a fair and generalizable model, we combine equations 1 and 2 and solve:

$$\hat{u}, \hat{w}, \hat{v} = \arg \min_{u,w,v} \mathcal{L}_{sl} + \alpha \mathcal{L}_{fair}^{s}, \tag{3}$$

to learn extracting features that are discriminative for performing the original classification task via $g_v(\cdot)$. The high-level description of this procedure is presented in Figure 1, top portion.

The above approach would suffice in practice if we have a single source domain, i.e., the data distribution is stationery and the testing data points are drawn from the source domain distribution. In our formulation, we consider that the test data is drawn from a second target domain $\mathcal{T}$ with a different data distribution $P_{\mathcal{T}}(\boldsymbol{X}, \boldsymbol{A}) \neq P_{\mathcal{S}}(\boldsymbol{X}, \boldsymbol{A})$. The target domain may be result of drifts in the input space or can occur when we want to use the model in a different domain. We also assume that we only have access to the unannotated dataset $\mathcal{D}^t = (X^t, A^t)$ in the target domain. Due to the distribution gap between the two domains, we need to update the model to remain fair in the target domain which will require annotating $\mathcal{D}^t$. Our goal is to make this process more practical by relaxing the need for data annotation. To this end, we formulate this problem in a UDA setting. UDA tackles the challenge of performance degradation under domain shift. The core idea in UDA is to improve generalization on the target domain via updating the encoder network such that the empirical distance between the distributions $e_u(P_{\mathcal{S}}(\boldsymbol{X}, \boldsymbol{A}))$ and $e_u(P_{\mathcal{T}}(\boldsymbol{X}, \boldsymbol{A}))$ is minimized, i.e., the two distributions are aligned such that the embedding space becomes domain agnostic. Under this restriction, the classifier $g_v(\cdot)$ that is trained on the source domain will generalize on the target domain. While this idea has been explored extensively in the UDA literature, it is insufficient to guarantee fairness after the adaptation phase. Our goal is to extend UDA to perserve model fairness in the target domain in addition to maintaining model generalization.

## 4 Proposed Algorithm

While adversarial learning has been used extensively to address UDA similar to training a fair model, solving two coupled adversarial learning problems to address our problem can be challenging. In our approach we still use adversarial learning to preserve fairness but benefit from metric learning to maintain model generalization Lee et al. (2019); Redko et al. (2017). The block-diagram description of our proposed approach is presented in Figure 1. We follow a two phase process. Initially, we train a fair model on the source domain dataset $(X^s, A^s, Y^s)$ and then update it to work well on the target domain. To train a fair model, we use the following three steps iteratively to solve equation 3:

1. We optimize the classifier $f_\theta(\cdot) = (g_v \circ e_u)(\cdot)$ network in an end-to-end scheme by minimizing equation 1. This process will generate informative and discriminative latent features for decision making.

2. We then fix the feature extractor encoder $e_u(\cdot)$ and optimize the sensitive attribute classifier $h_w(\cdot)$ by minimizing the loss in equation 2. This step will enforce the sensitive attribute classifier to extract information from the representations in the embedding space that can be used for predicting the sensitive attribute $A$.

3. We freeze the sensitive attribute classifier $h_w(\cdot)$ and update the encoder subnetwork $e_u(\cdot)$ in order to maximize the fairness loss function in equation 2. This step will force the encoder to produce representations that are independent from the sensitive attribute $A$ to enforce fairness.

The above steps leads to training a fair and generalizable model. In the second phase, we update the model to remain fair and generalizable when used on the target domain. We first explain the classic UDA approach.

The classic adaptation process relies only on aligning the two distributions in the embedding space, i.e., $e(P_\mathcal{S}(\boldsymbol{X}, \boldsymbol{A})) \approx e(P_\mathcal{T}(\boldsymbol{X}, \boldsymbol{A}))$. We follow metric minimization to enforce domain alignment Lee et al. (2019); Redko et al. (2017). The idea is to select a suitable probability distribution distance $d(\cdot, \cdot)$ and minimize it as a loss function at the encoder output, i.e. $d(e(P_\mathcal{S}(\boldsymbol{X}, \boldsymbol{A})), e(P_\mathcal{T}(\boldsymbol{X}, \boldsymbol{A})))$. As a result, the encoder is trained to guarantee domain-agnostic embedding features at its output. Compared to using adversarial learning, this approach requires less hyperparameter tuning and the resulting optimization problem is more stable. The choice of the distribution distance $d(\cdot, \cdot)$ is a design choice and various metric have been used for this purpose. We use the Sliced Wasserstein Distance (SWD) Redko et al. (2017) for this purpose. SWD is defined based on optimal transport or the Wasserstein Distance (WD) metric to broaden its applicability in deep learning. The upside of using WD is that it has a non-zero gradient even when the support for two distributions are non-overlapping. WD has been used successfully to address UDA but the downside of using WD is that it is defined in terms of an optimization problem. As a result, minimizing WD directly is a challenging task because often we need to solve another optimization problem to compute WD. The idea behind defining SWD is to develop a metric with closed-form solution by slicing two high-dimensional distributions to generate 1D projected distributions. Since WD has a closed-form solution for $1D$ distributions, SWD between the two high-dimensional distributions is computed as the average of these $1D$ WD slices. In addition to having a closed-form solution, SWD can be computed using empirical samples from the two distributions as follows:

$$\mathcal{L}_{swd} = \frac{1}{K} \sum_{i=1}^{K} WD^1(\langle e(\boldsymbol{x}^s, \boldsymbol{a}^s), \gamma_i \rangle, \langle e(\boldsymbol{x}^t, \boldsymbol{a}^t), \gamma_i \rangle), \tag{4}$$

where, $WD^1(\cdot, \cdot)$ denotes the $1D$ WD distance, $K$ is the number of random 1D projections we are averaging over and $\gamma_i$ is one such projection direction. We use random projection to estimate averaging over all possible projections. We can then solve the following problem to maintain model generalization on the source domain:

$$\mathcal{L}_{sl} + \gamma \mathcal{L}_{swd}. \tag{5}$$

If we only align the two distributions using equation 5, the model fairness can be compromised because when the encoder is updated to maintain model generalization, there is no guarantee that the embedding space remains independent from the sensitive attributes. Hence, the model can become biased. To preserve fairness in the target domain under distributional shifts, we augment the iterative steps $(1) - (3)$ described above with the following two steps:

4. We minimize the empirical SWD distance between $e(P_\mathcal{S}(\boldsymbol{X}, \boldsymbol{A}))$ and $e(P_\mathcal{T}(\boldsymbol{X}, \boldsymbol{A}))$ via equation 4. This step ensures the source-trained classifier $g(\cdot)$ will generalize on the target domain samples from $e(P_\mathcal{T}(\boldsymbol{X}, \boldsymbol{A}))$.

5. We repeat steps (2) and (3) using solely the sensitive attributes of the target domain.

The additional steps will update the model on the target domain to preserve both fairness and generalization accuracy. Following steps (1)-(5), the total loss function that we minimize would become:

$$\mathcal{L}_{bce}(\hat{y}, y_s) + \alpha \mathcal{L}_{fair}^s + \beta \mathcal{L}_{fair}^t + \gamma \mathcal{L}_{swd}, \tag{6}$$

where the trade-off hyperparameters $\alpha, \beta$, and $\gamma$ can be tuned using cross validation. Algorithm 1 summarizes the above described training process for our proposed algorithm, named *FairAdapt*.

## 5 Empirical Validation

We adopt existing common datasets in the AI fairness literature and tailor them for our formulation.

### 5.1 Experimental Setup

We first describe our empirical exploration setting.

#### 5.1.1 Datasets and Tasks

Common datasets in the fairness literature pose binary decision-making problems, e.g., approval of a credit application, alongside relevant features used for decision-making by professionals, e.g., employment history, credit history etc., and group-related sensitive attributes, e.g., sex, race, nationality, etc. Based on sensitive group membership, data points can be part of privileged or unprivileged subgroups. For example, with respect to sex, men are part of the privileged group while women are part of the unprivileged group. We perform experiments on three datasets widely used by the AI fairness community. We consider *sex* as our sensitive attribute because it is recorded for all three datasets. These datasets are:

> **Algorithm 1** FairAdapt $(\alpha, \beta, \gamma, thresh, ITR)$
> 1: **for** $itr = 1, \dots, ITR$ **do**
> 2:     **Source Training**:
> 3:     Optimize $\alpha\mathcal{L}_{bce}$ via 1.
> 4:     Optimize $\beta\mathcal{L}_{fair}$ via 2 and freezing $u$.
> 5:     Optimize $-\beta\mathcal{L}_{fair}$ via 2 and freezing $h$.
> 6:     **if** $itr > thresh$ **then**
> 7:         **Target Adaptation**:
> 8:         Optimize $\gamma\mathcal{L}_{swd}$ via 4.
> 9:         Optimize $\beta\mathcal{L}_{fair}$ via 2 and freezing $u$.
> 10:        Optimize $-\beta\mathcal{L}_{fair}$ via 2 and freezing $h$.
> 11:     **end if**
> 12: **end for**
> 13: **return** $u, g$

The **UCI Adult dataset**[1] is part of the UCI database Dua & Graff (2017) and consists of 1994 US Census data. The task associated with the dataset is predicting whether annual income exceeds 50k. After data cleaning, the dataset consists of more than $48,000$ entries. Possible sensitive attributes for this dataset include *sex* and *race*.

The **UCI German credit dataset** [2] contains financial information for 1000 different people applying for credit and is also part of the UCI database. The predictive task involves categorizing individuals as acceptable or non-acceptable credit risks. *Sex* and *age* are possible sensitive attributes for the German dataset.

The **Correctional Offender Management Profiling for Alternative Sanctions (COMPAS) recidivism dataset** [3] maintains information of over $5,000$ individuals' criminal records. Models trained on this dataset attempt to predict people's two year risk of recidivism. For the COMPAS dataset, *sex* and *race* may be used as sensitive attributes.

#### 5.1.2 Evaluation Protocol

Experiments on these datasets have primarily considered random 70/30 splits for the training and test splits. While such data splits are useful in evaluating overfitting for fairness algorithms, features for training and test sets will be sampled from the same data distribution. As a result, randomly splitting the datasets is not suitable for our learning setting because domain shift will not exist between the training and the testing splits. Instead, we consider natural data splits obtained from sub-sampling the three datasets along different criteria to generate the training and testing splits. We show that compared to random splits, where learning a model that guarantees fairness on the source domain is often enough to guarantee fairness on the target domain predictions, domain discrepancy between the source and target domains can lead to biased or degenerate predictions on the target domain, even if the model is initially trained

---

[1] https://archive.ics.uci.edu/ml/datasets/Adult
[2] https://archive.ics.uci.edu/ml/datasets/statlog+(German+credit+data)
[3] https://github.com/propublica/COMPAS-analysis/

to be fair. For details about the splits for each dataset, please refer to the supplementary material. In short, these splits introduce domain gap between the testing and training splits to generate appropriate tasks for our setting.

Next, for each of the three datasets, we will generate source/target data splits where ignoring domain discrepancy between the source and target can negatively impact model fairness. Per dataset, we produce three such splits. We characterize the label distributions and sensitive attribute conditional distributions for the Adult dataset in Table 1. We provide similar analysis for the German and COMPAS datasets in the supplementary material.

**Adult Dataset**. We use age, education and race to generate the source and target domains. These domains can be a natural occurrence in practice, as gathered census information may differ along these axes geographically. For example, urban population is on average more educated than rural population [4], and more ethnically diverse [5]. Thus, a fair model trained on one of the two populations will need to overcome distribution shift when evaluated on the other population. The source/target splits we consider are as follows:

1. **Source Domain:** White, +12 education years. **Target Domain:** Non-white, Less than 12 education years.

2. **Source Domain:** White, Older than 30. **Target Domain:** Non-white, younger than 40.

3. **Source Domain:** Younger than 70, +12 ed. years. **Target Domain:** Older than 70, less than 12 ed. years.

In Table 1, we analyze the conditional distributions of the labels and sensitive attribute for the above data splits. For the random split (A), we see that the conditional distributions of the sensitive attributes are identical in both domains which is expected due to absence of domain shift. For the three splits that we generated, we observe all three distributions: $P(Y), P(A|Y=0), P(A|Y=1)$ differ between the source and the target domains. We also note the label distribution becomes more skewed towards $Y = 0$. Common UDA methods rely on establishing a shared embedding space for both the source and target distributions. These approaches typically prioritize domain-invariance and are agnostic to sensitive attribute conditional probabilities necessary for maintaining prediction fairness. Hence, based on the probability landscape showcased in Table 1, such methods may not be suitable for preserving fairness.

| Split | Source | | | | Target | | | |
|-------|--------|------|--------|--------|--------|------|--------|--------|
| | Size | Y=0 | A=0\|Y=0 | A=0\|Y=1 | Size | Y=0 | A=0\|Y=0 | A=0\|Y=1 |
| A | 34120 | 0.76 | 0.39 | 0.15 | 14722 | 0.76 | 0.39 | 0.15 |
| A1 | 12024 | 0.53 | 0.41 | 0.16 | 5393 | 0.91 | 0.49 | 0.18 |
| A2 | 29466 | 0.66 | 0.34 | 0.14 | 2219 | 0.97 | 0.48 | 0.30 |
| A3 | 11887 | 0.52 | 0.42 | 0.16 | 778 | 0.89 | 0.39 | 0.17 |

Table 1: Data split statistics corresponding to the Adult dataset: the row with no number, i.e., "A", corresponds to random data splits. The numbered rows, i.e., A1,A2,A3 correspond to statistics for specific splits that we prepared. The columns represent the probabilities of specific outcomes for specific splits, e.g., $P(Y = 0)$, when using *sex* as sensitive attribute.

### 5.1.3 Fairness Metrics

There exist a multitude of criteria developed for evaluating algorithmic fairness Mehrabi et al. (2021). The goal is to define fairness intuitively and then come up with a computable quantitative metric based on a notion of independence. In the context of datasets presenting a privileged and unprivileged group, these metrics rely on ensuring predictive parity between the two groups under different constraints. The most common fairness metric employed is demographic parity (DP) $P(\hat{Y} = 1|A = 0) = P(\hat{Y} = 1|A = 1)$, which is optimized when predicted label probability is identical across the two groups. However, DP only ensures similar representation between the two groups, while ignoring actual label distribution. Equal opportunity (EO) Hardt et al. (2016) conditions the fairness value on the true label $Y$, and is optimized when $P(\hat{Y} = 1|A = 0, Y = 1) = P(\hat{Y} = 1|A = 1, Y = 1)$. EO is preferred when the label distribution is different across privilege classes, i.e., $P(Y|A = 0) \neq P(Y|A = 1)$. A more constrained fairness metric is averaged odds (AO), which is minimized when outcomes are the same conditioned on both labels and sensitive attributes, i.e., $P(\hat{Y}|A = 0, Y = y) = P(\hat{Y}|A = 1, Y = y), y \in \{0, 1\}$. EO is a special case of AO, for the case where $y = 1$. Following the AI fairness literature, we report the " left hand side and right hand side difference $\Delta$" for each of the

---
[4]https://www.ers.usda.gov/topics/rural-economy-population/employment-education/rural-education/
[5]https://www.ers.usda.gov/data-products/chart-gallery/gallery/chart-detail/?chartId=99538

above measures. Under this format, $\Delta$ values that are close to $0$ will signify that the model maintains fairness, while values close to $1$ signify a lack of fairness. Tuning a model to optimize fairness may incur accuracy trade offs Madras et al. (2018a); Kleinberg et al. (2016); Wick et al. (2019). For example, a classifier which predicts every element to be part of the same group, e.g., $P(\hat{Y} = 0) = 1$ will obtain $\Delta EO = \Delta EO = \Delta AO = 0$, without providing informative predictions. Our approach has the advantage that the regularizers of the three employed losses $\mathcal{L}_{CE}, \mathcal{L}_{fair}, \mathcal{L}_{swd}$ can be tuned in accordance with the importance of accuracy against fairness for a specific task.

### 5.1.4 Methods for Comparison

To the best of our knowledge, no prior method has exactly addressed our learning setting. To offer extensive evaluation, we compare our work against seven fairness preserving algorithms: Meta-Algorithm for Fair Classification (MC) Celis et al. (2019a), Adversarial Debiasing (AD) Zhang et al. (2018), Reject Option Classification (ROC) Kamiran et al. (2012), Exponentiated Gradient Reduction (EGR) Agarwal et al. (2018), Learning Fair Representations (LFR) Zemel et al. (2013), Calibrated Equal Odds (CEO) Pleiss et al. (2017), Reweighing Pre-processing (RP) Kamiran & Calders (2012). Implementations for these algorithms are available in the AIF360 Bellamy et al. (2018) package. Results reveal the superiority of our approach when distributional shift is present between source and target. We additionally report as baseline (Base) minimizing only $\mathcal{L}_{bce}$ without optimizing fairness or distributional distance. This baseline corresponds to the performance of a naive source-trained classifier and serves as a lower bound.

### 5.2 Comparison Results

We report balanced accuracy (Acc.), demographic parity ($\Delta DP$), equalized odds ($\Delta EO$) and averaged opportunity ($\Delta AO$) in our comparison results to study both accuracy and fairness. Desirable accuracy values are close to $1$, while desirable fairness metric values should be close to $0$. Prior studies have shown that there is a trade-off between the performance accuracy and the model fairness. Results are averaged over 10 runs to make comparisons statistically meaningful. We use *sex* as the sensitive attribute $A$, as it is shared across all datasets.

| Alg. | Adult | | | | German | | | | COMPAS | | | |
|---|---|---|---|---|---|---|---|---|---|---|---|---|
| | Acc. | $\Delta DP$ | $\Delta EO$ | $\Delta AO$ | Acc. | $\Delta DP$ | $\Delta EO$ | $\Delta AO$ | Acc. | $\Delta DP$ | $\Delta EO$ | $\Delta AO$ |
| Base | $0.74^{\pm0.00}$ | $0.37^{\pm0.00}$ | $0.38^{\pm0.01}$ | $0.33^{\pm0.01}$ | $0.67^{\pm0.01}$ | $0.22^{\pm0.07}$ | $0.14^{\pm0.02}$ | $0.18^{\pm0.05}$ | $0.68^{\pm0.00}$ | $0.32^{\pm0.02}$ | $0.42^{\pm0.03}$ | $0.30^{\pm0.02}$ |
| MC | $0.71^{\pm0.01}$ | $0.13^{\pm0.07}$ | $0.10^{\pm0.07}$ | $0.10^{\pm0.07}$ | $0.66^{\pm0.01}$ | $0.07^{\pm0.04}$ | $0.04^{\pm0.02}$ | $0.05^{\pm0.03}$ | $0.65^{\pm0.01}$ | $0.18^{\pm0.08}$ | $0.16^{\pm0.11}$ | $0.14^{\pm0.08}$ |
| AD | $0.66^{\pm0.02}$ | $0.09^{\pm0.03}$ | $0.10^{\pm0.09}$ | $0.06^{\pm0.05}$ | $0.52^{\pm0.00}$ | $0.55^{\pm0.34}$ | $0.62^{\pm0.30}$ | $0.57^{\pm0.33}$ | $0.64^{\pm0.03}$ | $0.19^{\pm0.22}$ | $0.19^{\pm0.19}$ | $0.21^{\pm0.22}$ |
| ROC | $0.71^{\pm0.00}$ | $0.01^{\pm0.00}$ | $0.05^{\pm0.00}$ | $0.06^{\pm0.00}$ | $0.66^{\pm0.00}$ | $0.05^{\pm0.00}$ | $0.07^{\pm0.00}$ | $0.03^{\pm0.00}$ | $0.52^{\pm0.00}$ | $0.02^{\pm0.00}$ | $0.05^{\pm0.00}$ | $0.02^{\pm0.00}$ |
| EGR | $0.65^{\pm0.00}$ | $0.05^{\pm0.00}$ | $0.01^{\pm0.01}$ | $0.01^{\pm0.00}$ | $0.55^{\pm0.02}$ | $0.02^{\pm0.02}$ | $0.07^{\pm0.03}$ | $0.03^{\pm0.02}$ | $0.65^{\pm0.01}$ | $0.09^{\pm0.02}$ | $0.04^{\pm0.02}$ | $0.04^{\pm0.02}$ |
| LFR | $0.71^{\pm0.01}$ | $0.02^{\pm0.02}$ | $0.05^{\pm0.02}$ | $0.06^{\pm0.02}$ | $0.65^{\pm0.01}$ | $0.01^{\pm0.01}$ | $0.09^{\pm0.02}$ | $0.02^{\pm0.01}$ | $0.65^{\pm0.01}$ | $0.01^{\pm0.00}$ | $0.04^{\pm0.01}$ | $0.04^{\pm0.01}$ |
| CEO | $0.68^{\pm0.00}$ | $0.06^{\pm0.00}$ | $0.02^{\pm0.00}$ | $0.01^{\pm0.00}$ | $0.61^{\pm0.03}$ | $0.03^{\pm0.01}$ | $0.09^{\pm0.04}$ | $0.03^{\pm0.02}$ | $0.64^{\pm0.01}$ | $0.11^{\pm0.01}$ | $0.11^{\pm0.01}$ | $0.07^{\pm0.01}$ |
| RP | $0.71^{\pm0.00}$ | $0.01^{\pm0.00}$ | $0.05^{\pm0.00}$ | $0.06^{\pm0.00}$ | $0.65^{\pm0.00}$ | $0.00^{\pm0.00}$ | $0.08^{\pm0.00}$ | $0.01^{\pm0.00}$ | $0.66^{\pm0.00}$ | $0.01^{\pm0.00}$ | $0.05^{\pm0.00}$ | $0.02^{\pm0.00}$ |
| Ours | $0.71^{\pm0.00}$ | $0.00^{\pm0.00}$ | $0.05^{\pm0.00}$ | $0.07^{\pm0.00}$ | $0.68^{\pm0.01}$ | $0.01^{\pm0.00}$ | $0.03^{\pm0.02}$ | $0.03^{\pm0.01}$ | $0.67^{\pm0.00}$ | $0.00^{\pm0.00}$ | $0.05^{\pm0.01}$ | $0.03^{\pm0.00}$ |

Table 2: Results for random data splits.

We first report performance results for standard random splits that are commonly used in the fairness literature in Table 2. Since for standard splits, the source and the target are sampled from the same distribution, there is no domain shift. We observe the baseline approach obtains highest or close to highest accuracy across datasets, but does not lead to fair predictions according to the three fairness metrics. The rest of the methods preserve fairness significantly better than the baseline but their performance accuracy values are less than the baseline. This observation aligns with what has been reported in the fairness literature. Importantly, our method leads to best accuracy performance amongst the fairness preserving methods while also leading to minimum demographic parity on the Adult and COMPAS datasets, which indicates that the embedding space is fully independent from the sensitive attributes. We also see that our method achieves best equalized odds difference on the German dataset, as well as close to best average opportunity difference on the German and COMPAS datasets, despite the fact that our method is not directly minimizing these metrics. We conclude that our algorithm successfully learns a competitively fair model when domain shift does not exist while leading to the best performance accuracy compared to fairness preserving methods.

We next present results for the three data splits for each of the considered datasets that we prepared. These are custom splits for each dataset such that domain shift exists during the testing phase.

**Adult dataset** We report results on the three splits of the Adult dataset in Table 3.

We first note that out of the considered methods, our approach is the only one capable of maintaining both fairness and competitive accuracy on all data splits. On the first split, MC obtains the highest accuracy of $0.66$, however is not able to maintain fairness. LFR, CEO and RP are able to maintain fairness, which is matched by our method on the demographic parity metric. On the second split, we are able to obtain best fairness results with respect to all fairness metrics: $\Delta DP$, $\Delta EO$, $\Delta AO$. In contrast, all other fairness methods are unable to offer competitive performance for $\Delta EO$, $\Delta AO$. On the third split, CEO and ROC produce degenerate results. Out of the remaining methods, we are able to once again obtain best fairness scores with respect to all metrics. Similar performance is obtained on the third split. We conclude that existing fairness-preserving methods struggle with domain shift between the source and target, while our method is positioned to overcome the challenge of domain shift.

| Alg. | Race, Education | | | | Race, Age | | | | Age, Education | | | |
|---|---|---|---|---|---|---|---|---|---|---|---|---|
| | Acc. | $\Delta DP$ | $\Delta EO$ | $\Delta AO$ | Acc. | $\Delta DP$ | $\Delta EO$ | $\Delta AO$ | Acc. | $\Delta DP$ | $\Delta EO$ | $\Delta AO$ |
| Base | $0.66^{\pm0.06}$ | $0.40^{\pm0.19}$ | $0.61^{\pm0.25}$ | $0.49^{\pm0.21}$ | $0.60^{\pm0.02}$ | $0.18^{\pm0.10}$ | $0.23^{\pm0.12}$ | $0.20^{\pm0.10}$ | $0.62^{\pm0.03}$ | $0.81^{\pm0.25}$ | $0.85^{\pm0.28}$ | $0.83^{\pm0.27}$ |
| MC | $0.66^{\pm0.02}$ | $0.19^{\pm0.14}$ | $0.27^{\pm0.20}$ | $0.21^{\pm0.17}$ | $0.64^{\pm0.02}$ | $0.12^{\pm0.14}$ | $0.19^{\pm0.13}$ | $0.15^{\pm0.11}$ | $0.62^{\pm0.03}$ | $0.81^{\pm0.27}$ | $0.85^{\pm0.27}$ | $0.83^{\pm0.27}$ |
| AD | $0.64^{\pm0.04}$ | $0.15^{\pm0.13}$ | $0.24^{\pm0.20}$ | $0.18^{\pm0.16}$ | $0.60^{\pm0.04}$ | $0.21^{\pm0.17}$ | $0.22^{\pm0.10}$ | $0.21^{\pm0.09}$ | $0.59^{\pm0.04}$ | $0.64^{\pm0.25}$ | $0.69^{\pm0.25}$ | $0.67^{\pm0.25}$ |
| ROC | $0.56^{\pm0.00}$ | $0.40^{\pm0.00}$ | $0.32^{\pm0.00}$ | $0.38^{\pm0.00}$ | $0.64^{\pm0.00}$ | $0.02^{\pm0.00}$ | $0.15^{\pm0.00}$ | $0.09^{\pm0.00}$ | $0.50^{\pm0.00}$ | $0.00^{\pm0.00}$ | $0.00^{\pm0.00}$ | $0.00^{\pm0.00}$ |
| EGR | $0.64^{\pm0.00}$ | $0.12^{\pm0.01}$ | $0.26^{\pm0.03}$ | $0.17^{\pm0.01}$ | $0.61^{\pm0.01}$ | $0.01^{\pm0.00}$ | $0.14^{\pm0.06}$ | $0.07^{\pm0.03}$ | $0.54^{\pm0.03}$ | $0.24^{\pm0.02}$ | $0.25^{\pm0.14}$ | $0.24^{\pm0.07}$ |
| LFR | $0.63^{\pm0.01}$ | $0.02^{\pm0.01}$ | $0.04^{\pm0.01}$ | $0.01^{\pm0.00}$ | $0.63^{\pm0.02}$ | $0.04^{\pm0.01}$ | $0.24^{\pm0.04}$ | $0.14^{\pm0.02}$ | $0.53^{\pm0.01}$ | $0.01^{\pm0.00}$ | $0.04^{\pm0.01}$ | $0.02^{\pm0.00}$ |
| CEO | $0.64^{\pm0.00}$ | $0.00^{\pm0.00}$ | $0.05^{\pm0.00}$ | $0.01^{\pm0.00}$ | $0.62^{\pm0.00}$ | $0.04^{\pm0.00}$ | $0.24^{\pm0.00}$ | $0.14^{\pm0.00}$ | $0.50^{\pm0.00}$ | $0.00^{\pm0.00}$ | $0.00^{\pm0.00}$ | $0.00^{\pm0.00}$ |
| RP | $0.63^{\pm0.00}$ | $0.00^{\pm0.00}$ | $0.00^{\pm0.00}$ | $0.01^{\pm0.00}$ | $0.65^{\pm0.00}$ | $0.02^{\pm0.00}$ | $0.28^{\pm0.00}$ | $0.15^{\pm0.00}$ | $0.53^{\pm0.00}$ | $0.01^{\pm0.00}$ | $0.04^{\pm0.00}$ | $0.02^{\pm0.00}$ |
| Ours | $0.62^{\pm0.01}$ | $0.00^{\pm0.00}$ | $0.02^{\pm0.01}$ | $0.02^{\pm0.01}$ | $0.58^{\pm0.01}$ | $0.01^{\pm0.00}$ | $0.05^{\pm0.05}$ | $0.03^{\pm0.02}$ | $0.52^{\pm0.01}$ | $0.01^{\pm0.01}$ | $0.02^{\pm0.02}$ | $0.01^{\pm0.01}$ |

Table 3: Performance results for the three splits of the Adult dataset

**COMPAS dataset** results for the COMPAS dataset are reported in Table 4.

On the first data split, our method is able to obtain the best fairness performance with respect to all three metrics. RP achieves best accuracy, however comes second in terms of fairness performance. On the second split, several methods produce degenerate solutions, such as RP, CEO, LFR or ROC. A degenerate solution is undesirable, as fairness is minimized by assigning the same label to all samples. In contrast, our method is strikes a balance between accuracy and fairness. On the third split, FairAdapt achieves best results both in accuracy and fairness. Again, several methods produce degenerate solutions. AD matches our performance with respect to accuracy, but fails to maintain fairness. We conclude FairAdapt is effective on COMPAS, as it maintains both accuracy and fairness under domain shift.

| Alg. | Age, Priors | | | | Race, Age, Priors | | | | Age, Priors, Charge | | | |
|---|---|---|---|---|---|---|---|---|---|---|---|---|
| | Acc. | $\Delta DP$ | $\Delta EO$ | $\Delta AO$ | Acc. | $\Delta DP$ | $\Delta EO$ | $\Delta AO$ | Acc. | $\Delta DP$ | $\Delta EO$ | $\Delta AO$ |
| Base | $0.58^{\pm0.03}$ | $0.33^{\pm0.09}$ | $0.35^{\pm0.07}$ | $0.33^{\pm0.08}$ | $0.59^{\pm0.04}$ | $0.51^{\pm0.26}$ | $0.60^{\pm0.35}$ | $0.55^{\pm0.27}$ | $0.60^{\pm0.03}$ | $0.56^{\pm0.13}$ | $0.60^{\pm0.20}$ | $0.56^{\pm0.14}$ |
| MC | $0.60^{\pm0.02}$ | $0.30^{\pm0.15}$ | $0.33^{\pm0.23}$ | $0.30^{\pm0.17}$ | $0.50^{\pm0.00}$ | $0.00^{\pm0.00}$ | $0.00^{\pm0.00}$ | $0.00^{\pm0.00}$ | $0.53^{\pm0.02}$ | $0.33^{\pm0.21}$ | $0.33^{\pm0.23}$ | $0.33^{\pm0.21}$ |
| AD | $0.58^{\pm0.05}$ | $0.72^{\pm0.28}$ | $0.82^{\pm0.22}$ | $0.75^{\pm0.26}$ | $0.61^{\pm0.02}$ | $0.77^{\pm0.31}$ | $0.83^{\pm0.23}$ | $0.79^{\pm0.27}$ | $0.57^{\pm0.01}$ | $0.86^{\pm0.14}$ | $0.86^{\pm0.14}$ | $0.86^{\pm0.14}$ |
| ROC | $0.50^{\pm0.00}$ | $0.00^{\pm0.00}$ | $0.00^{\pm0.00}$ | $0.00^{\pm0.00}$ | $0.50^{\pm0.00}$ | $0.00^{\pm0.00}$ | $0.00^{\pm0.00}$ | $0.00^{\pm0.00}$ | $0.50^{\pm0.00}$ | $0.00^{\pm0.00}$ | $0.00^{\pm0.00}$ | $0.00^{\pm0.00}$ |
| EGR | $0.51^{\pm0.00}$ | $0.12^{\pm0.04}$ | $0.06^{\pm0.04}$ | $0.08^{\pm0.05}$ | $0.52^{\pm0.02}$ | $0.10^{\pm0.01}$ | $0.05^{\pm0.04}$ | $0.09^{\pm0.01}$ | $0.54^{\pm0.02}$ | $0.08^{\pm0.06}$ | $0.09^{\pm0.08}$ | $0.08^{\pm0.05}$ |
| LFR | $0.59^{\pm0.02}$ | $0.02^{\pm0.01}$ | $0.07^{\pm0.04}$ | $0.04^{\pm0.02}$ | $0.50^{\pm0.00}$ | $0.00^{\pm0.00}$ | $0.00^{\pm0.00}$ | $0.00^{\pm0.00}$ | $0.50^{\pm0.00}$ | $0.00^{\pm0.00}$ | $0.00^{\pm0.00}$ | $0.00^{\pm0.00}$ |
| CEO | $0.50^{\pm0.00}$ | $0.00^{\pm0.00}$ | $0.00^{\pm0.00}$ | $0.00^{\pm0.00}$ | $0.50^{\pm0.00}$ | $0.00^{\pm0.00}$ | $0.00^{\pm0.00}$ | $0.00^{\pm0.00}$ | $0.50^{\pm0.00}$ | $0.00^{\pm0.00}$ | $0.00^{\pm0.00}$ | $0.00^{\pm0.00}$ |
| RP | $0.61^{\pm0.00}$ | $0.02^{\pm0.00}$ | $0.05^{\pm0.00}$ | $0.04^{\pm0.00}$ | $0.50^{\pm0.00}$ | $0.00^{\pm0.00}$ | $0.00^{\pm0.00}$ | $0.00^{\pm0.00}$ | $0.50^{\pm0.00}$ | $0.00^{\pm0.00}$ | $0.00^{\pm0.00}$ | $0.00^{\pm0.00}$ |
| Ours | $0.58^{\pm0.01}$ | $0.01^{\pm0.01}$ | $0.02^{\pm0.01}$ | $0.01^{\pm0.01}$ | $0.56^{\pm0.03}$ | $0.19^{\pm0.05}$ | $0.35^{\pm0.17}$ | $0.28^{\pm0.08}$ | $0.57^{\pm0.00}$ | $0.02^{\pm0.00}$ | $0.01^{\pm0.00}$ | $0.02^{\pm0.00}$ |

Table 4: Performance results for the three splits of the COMPAS dataset

**German dataset** in Table 5, we present the results on the German dataset.

In the first data split, our approach achieves best performance in terms of accuracy while obtaining a close to optimal demographic parity value. This highlights the ability of our method to strike a balance between accuracy and fairness, making it a compelling choice for domain adaptation tasks. Moving on to the second data split, our method achieves competitive performance for accuracy, and close to best performance for two of the three fairness metrics. On the last data split, our method outperforms all other algorithms that do not produce degenerate results (all approaches besides ROC, CEO) in terms of accuracy and both demographic parity and averaged opportunity. This proves the robustness of our approach, even in challenging scenarios, where fairness is a critical concern.

| Alg. | Employment | | | | Credit hist., Empl. | | | | Credit hist., Empl. | | | |
|---|---|---|---|---|---|---|---|---|---|---|---|---|
| | Acc. | $\Delta DP$ | $\Delta EO$ | $\Delta AO$ | Acc. | $\Delta DP$ | $\Delta EO$ | $\Delta AO$ | Acc. | $\Delta DP$ | $\Delta EO$ | $\Delta AO$ |
| Base | $0.61^{\pm0.05}$ | $0.08^{\pm0.09}$ | $0.06^{\pm0.05}$ | $0.07^{\pm0.07}$ | $0.59^{\pm0.02}$ | $0.26^{\pm0.23}$ | $0.32^{\pm0.26}$ | $0.27^{\pm0.25}$ | $0.55^{\pm0.02}$ | $0.30^{\pm0.27}$ | $0.20^{\pm0.20}$ | $0.25^{\pm0.25}$ |
| MC | $0.65^{\pm0.01}$ | $0.12^{\pm0.03}$ | $0.10^{\pm0.04}$ | $0.12^{\pm0.03}$ | $0.60^{\pm0.02}$ | $0.03^{\pm0.05}$ | $0.15^{\pm0.03}$ | $0.09^{\pm0.03}$ | $0.55^{\pm0.00}$ | $0.09^{\pm0.00}$ | $0.00^{\pm0.00}$ | $0.05^{\pm0.00}$ |
| AD | $0.53^{\pm0.02}$ | $0.63^{\pm0.23}$ | $0.70^{\pm0.24}$ | $0.65^{\pm0.21}$ | $0.54^{\pm0.04}$ | $0.41^{\pm0.31}$ | $0.47^{\pm0.27}$ | $0.44^{\pm0.27}$ | $0.53^{\pm0.01}$ | $0.56^{\pm0.22}$ | $0.57^{\pm0.30}$ | $0.57^{\pm0.24}$ |
| ROC | $0.54^{\pm0.00}$ | $0.14^{\pm0.00}$ | $0.05^{\pm0.00}$ | $0.11^{\pm0.00}$ | $0.51^{\pm0.00}$ | $0.18^{\pm0.00}$ | $0.50^{\pm0.00}$ | $0.33^{\pm0.00}$ | $0.50^{\pm0.00}$ | $0.00^{\pm0.00}$ | $0.00^{\pm0.00}$ | $0.00^{\pm0.00}$ |
| EGR | $0.58^{\pm0.01}$ | $0.28^{\pm0.02}$ | $0.43^{\pm0.04}$ | $0.33^{\pm0.03}$ | $0.52^{\pm0.01}$ | $0.18^{\pm0.05}$ | $0.58^{\pm0.28}$ | $0.36^{\pm0.15}$ | $0.51^{\pm0.01}$ | $0.59^{\pm0.03}$ | $0.67^{\pm0.00}$ | $0.62^{\pm0.02}$ |
| LFR | $0.65^{\pm0.02}$ | $0.01^{\pm0.02}$ | $0.04^{\pm0.02}$ | $0.02^{\pm0.02}$ | $0.63^{\pm0.03}$ | $0.00^{\pm0.01}$ | $0.11^{\pm0.06}$ | $0.07^{\pm0.02}$ | $0.53^{\pm0.00}$ | $0.02^{\pm0.00}$ | $0.08^{\pm0.00}$ | $0.05^{\pm0.00}$ |
| CEO | $0.50^{\pm0.00}$ | $0.00^{\pm0.00}$ | $0.00^{\pm0.00}$ | $0.00^{\pm0.00}$ | $0.61^{\pm0.00}$ | $0.00^{\pm0.00}$ | $0.16^{\pm0.00}$ | $0.08^{\pm0.00}$ | $0.50^{\pm0.00}$ | $0.00^{\pm0.00}$ | $0.00^{\pm0.00}$ | $0.00^{\pm0.00}$ |
| RP | $0.67^{\pm0.00}$ | $0.01^{\pm0.00}$ | $0.02^{\pm0.00}$ | $0.00^{\pm0.00}$ | $0.61^{\pm0.00}$ | $0.00^{\pm0.00}$ | $0.16^{\pm0.00}$ | $0.08^{\pm0.00}$ | $0.53^{\pm0.00}$ | $0.02^{\pm0.00}$ | $0.08^{\pm0.00}$ | $0.05^{\pm0.00}$ |
| Ours | $0.67^{\pm0.01}$ | $0.01^{\pm0.01}$ | $0.05^{\pm0.04}$ | $0.02^{\pm0.02}$ | $0.61^{\pm0.01}$ | $0.03^{\pm0.01}$ | $0.17^{\pm0.15}$ | $0.08^{\pm0.05}$ | $0.55^{\pm0.01}$ | $0.02^{\pm0.01}$ | $0.08^{\pm0.05}$ | $0.04^{\pm0.02}$ |

Table 5: Performance results for the three splits of the German dataset

From Tables 3–5, we conclude that algorithms for training fair models are vulnerable in our setting. FairAdapt is effective and well-suited for preserving model fairness and accuracy performance on tasks associated with domain shift. Our approach is the only algorithm out of the considered methods that is able to consistently achieve top performance both in terms of accuracy and fairness on nine data splits across three datasets. Its demonstrated robustness make it a promising choice for real-world applications where domain adaptation and fairness are crucial considerations.

## 5.3 Analytic and Ablative Experiments

To provide a more intuitive understanding of our method, we visualize the impact of domain shift by generating $2D$ embeddings of the source and target domain features in the shared embedding space. For this purpose, we employ the UMAP McInnes et al. (2020) visualization tool, which helps us create meaningful visual representations that encode the geometry of high dimensions. The resulting visualizations are presented in Figure 2. We have compared the source and target features resulting from a random split of the Adult dataset (Figure 2 (a)) with our first custom split (Figure 2 (b)). Upon examining the visualization of the random split, we notice that the source and target samples exhibit a considerable degree of similarity. However, when using a custom split, we observe a substantial discrepancy between the two distributions, indicating the existence of distributional mismatch. This disparity can have a significant impact on the model's ability to generalize effectively. Our numerical results align with this observation, indicating that in the presence of domain shift, maintaining both model generalization and fairness becomes a challenging task.

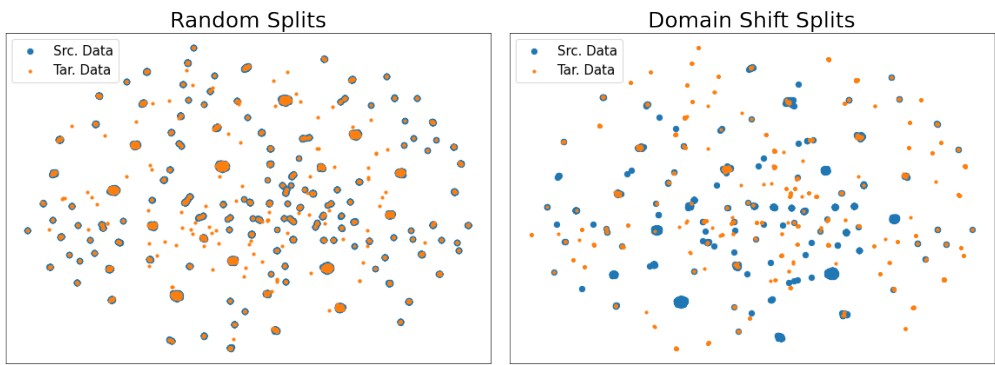

Figure 2: UMAP embeddings of the source and target feature spaces for random and custom splits of the Adult dataset

We additionally provide ablative experiments to investigate the impact of the different components of our approach on performance. In Table 6, we compare the performance on the COMPAS dataset of four variants of our algorithm: (1) *Base*, similar to the main experiments, where no fairness or distributional minimization metric is used, (2) *SWD*, only the loss $\mathcal{L}_{swd}$ is minimized (3) *Fair*, training is performed only with respect to $\mathcal{L}_{fair}$ on the source and target domains (4) Our complete pipeline using both fairness and adaptation objectives. We can see that on the first and third splits, utilizing all losses leads to the best performance in terms of fairness. On the second split, we are able to obtain highest balanced accuracy while improving $\Delta DP$ compared to only optimizing the SWD metric. In general, compared to the baseline and SWD variant, the Fair variant is able to achieve competitive fairness results at the cost of accuracy. The

SWD only approach achieves better accuracy but at the cost of fairness. Combining the two losses leads to improved accuracy over the Fair only model, and also improved fairness if accuracy is matched. Due to $\mathcal{L}_{swd}$ being minimized at the encoder output space, both classifier and fairness head benefit from a shared source-target feature space.

In the previous experiments, we only considered *sex* as the sensitive attribute. We assess the performance of our proposed algorithm when using a different sensitive attribute. For this purpose, we utilize the German dataset and designate *age* as the sensitive attribute. The results of these experiments are presented in Table 7. Similar to our experiments where *sex* was chosen as the sensitive attribute, FairAdapt continues to exhibit outstanding performance by achieving the best combined performance among all the methods. It achieves best or second best demographic parity scores on all splits, and maintains accuracy values close to the highest reported. This observation demonstrates the robustness of our approach in terms of the choice of sensitive attribute: our method can adapt to various fairness settings and has the potential to cover a wide range of domain adaptation tasks where fairness is a critical consideration.

| Alg. | Age, Priors | | | | Race, Age, Priors | | | | Age, Priors, Chrg. | | | |
|---|---|---|---|---|---|---|---|---|---|---|---|---|
| | Acc. | $\Delta DP$ | $\Delta EO$ | $\Delta AO$ | Acc. | $\Delta DP$ | $\Delta EO$ | $\Delta AO$ | Acc. | $\Delta DP$ | $\Delta EO$ | $\Delta AO$ |
| Base | $0.58^{\pm0.03}$ | $0.33^{\pm0.09}$ | $0.35^{\pm0.07}$ | $0.33^{\pm0.08}$ | $0.59^{\pm0.04}$ | $0.51^{\pm0.26}$ | $0.60^{\pm0.35}$ | $0.55^{\pm0.27}$ | $0.60^{\pm0.03}$ | $0.56^{\pm0.13}$ | $0.60^{\pm0.20}$ | $0.56^{\pm0.14}$ |
| SWD | $0.59^{\pm0.01}$ | $0.32^{\pm0.06}$ | $0.42^{\pm0.10}$ | $0.34^{\pm0.07}$ | $0.55^{\pm0.03}$ | $0.32^{\pm0.24}$ | $0.3^{\pm0.36}$ | $0.32^{\pm0.28}$ | $0.62^{\pm0.03}$ | $0.59^{\pm0.10}$ | $0.63^{\pm0.16}$ | $0.57^{\pm0.11}$ |
| Fair | $0.58^{\pm0.01}$ | $0.01^{\pm0.00}$ | $0.03^{\pm0.01}$ | $0.02^{\pm0.01}$ | $0.54^{\pm0.03}$ | $0.13^{\pm0.10}$ | $0.20^{\pm0.2}$ | $0.17^{\pm0.14}$ | $0.57^{\pm0.00}$ | $0.03^{\pm0.00}$ | $0.03^{\pm0.00}$ | $0.06^{\pm0.00}$ |
| Ours | $0.58^{\pm0.01}$ | $0.01^{\pm0.01}$ | $0.02^{\pm0.01}$ | $0.01^{\pm0.01}$ | $0.56^{\pm0.03}$ | $0.19^{\pm0.05}$ | $0.35^{\pm0.17}$ | $0.28^{\pm0.08}$ | $0.57^{\pm0.00}$ | $0.02^{\pm0.00}$ | $0.01^{\pm0.00}$ | $0.02^{\pm0.00}$ |

Table 6: Ablative experiments using a subset of losses on the COMPAS dataset

| Alg. | Empl. | | | | Credit hist., Empl. | | | | Credit hist., Empl. | | | |
|---|---|---|---|---|---|---|---|---|---|---|---|---|
| | Acc. | $\Delta DP$ | $\Delta EO$ | $\Delta AO$ | Acc. | $\Delta DP$ | $\Delta EO$ | $\Delta AO$ | Acc. | $\Delta DP$ | $\Delta EO$ | $\Delta AO$ |
| Base | $0.61^{\pm0.05}$ | $0.17^{\pm0.13}$ | $0.22^{\pm0.21}$ | $0.17^{\pm0.11}$ | $0.57^{\pm0.04}$ | $0.27^{\pm0.19}$ | $0.29^{\pm0.20}$ | $0.26^{\pm0.19}$ | $0.53^{\pm0.02}$ | $0.12^{\pm0.12}$ | $0.25^{\pm0.08}$ | $0.18^{\pm0.09}$ |
| MC | $0.67^{\pm0.01}$ | $0.31^{\pm0.12}$ | $0.17^{\pm0.05}$ | $0.25^{\pm0.10}$ | $0.59^{\pm0.00}$ | $0.21^{\pm0.00}$ | $0.16^{\pm0.00}$ | $0.19^{\pm0.00}$ | $0.55^{\pm0.01}$ | $0.27^{\pm0.19}$ | $0.30^{\pm0.20}$ | $0.27^{\pm0.19}$ |
| AD | $0.53^{\pm0.00}$ | $0.58^{\pm0.04}$ | $0.80^{\pm0.00}$ | $0.64^{\pm0.03}$ | $0.51^{\pm0.01}$ | $0.63^{\pm0.27}$ | $0.68^{\pm0.38}$ | $0.65^{\pm0.31}$ | $0.51^{\pm0.01}$ | $0.69^{\pm0.33}$ | $0.73^{\pm0.35}$ | $0.66^{\pm0.37}$ |
| ROC | $0.67^{\pm0.00}$ | $0.30^{\pm0.00}$ | $0.16^{\pm0.00}$ | $0.24^{\pm0.00}$ | $0.51^{\pm0.00}$ | $0.06^{\pm0.00}$ | $0.05^{\pm0.00}$ | $0.06^{\pm0.00}$ | $0.55^{\pm0.00}$ | $0.15^{\pm0.00}$ | $0.20^{\pm0.00}$ | $0.17^{\pm0.00}$ |
| EGR | $0.51^{\pm0.01}$ | $0.02^{\pm0.01}$ | $0.05^{\pm0.02}$ | $0.03^{\pm0.02}$ | $0.52^{\pm0.01}$ | $0.25^{\pm0.14}$ | $0.49^{\pm0.29}$ | $0.33^{\pm0.20}$ | $0.51^{\pm0.01}$ | $0.36^{\pm0.11}$ | $0.53^{\pm0.00}$ | $0.35^{\pm0.18}$ |
| LFR | $0.64^{\pm0.04}$ | $0.04^{\pm0.04}$ | $0.15^{\pm0.17}$ | $0.06^{\pm0.06}$ | $0.56^{\pm0.03}$ | $0.02^{\pm0.02}$ | $0.12^{\pm0.10}$ | $0.05^{\pm0.04}$ | $0.58^{\pm0.00}$ | $0.06^{\pm0.00}$ | $0.07^{\pm0.00}$ | $0.06^{\pm0.00}$ |
| CEO | $0.50^{\pm0.00}$ | $0.00^{\pm0.00}$ | $0.00^{\pm0.00}$ | $0.00^{\pm0.00}$ | $0.54^{\pm0.00}$ | $0.01^{\pm0.00}$ | $0.05^{\pm0.00}$ | $0.02^{\pm0.00}$ | $0.50^{\pm0.00}$ | $0.00^{\pm0.00}$ | $0.00^{\pm0.00}$ | $0.00^{\pm0.00}$ |
| RP | $0.60^{\pm0.00}$ | $0.00^{\pm0.00}$ | $0.13^{\pm0.00}$ | $0.05^{\pm0.00}$ | $0.54^{\pm0.00}$ | $0.01^{\pm0.00}$ | $0.05^{\pm0.00}$ | $0.02^{\pm0.00}$ | $0.50^{\pm0.00}$ | $0.00^{\pm0.00}$ | $0.00^{\pm0.00}$ | $0.00^{\pm0.00}$ |
| Ours | $0.66^{\pm0.01}$ | $0.01^{\pm0.00}$ | $0.08^{\pm0.02}$ | $0.04^{\pm0.01}$ | $0.57^{\pm0.01}$ | $0.00^{\pm0.00}$ | $0.06^{\pm0.02}$ | $0.03^{\pm0.00}$ | $0.55^{\pm0.00}$ | $0.00^{\pm0.00}$ | $0.31^{\pm0.00}$ | $0.14^{\pm0.00}$ |

Table 7: Results on the German dataset when optimizing fairness metrics with respect to the *age* sensitive attribute

For additional experiments about the dynamics of learning when our method is used, please refer to the Appendix. In summary, we analyzed the effect of adaptation process on target domain accuracy and demographic parity on the target domain as more training epochs are performed. We observed that the target accuracy consistently increased while demographic parity on both the source and target domains remained relatively unchanged, i.e., fairness is maintained. These observations validate that our algorithm leads to desired effects on the model performance.

# 6 Conclusions and Future Work

We study the problem of fairness under domain shift. Fairness preserving methods have overlooked the problem of domain shift when deploying a source trained model to a target domain. Our first contribution is providing different data splits for common datasets employed in fairness tasks which present significant domain shift between the source and target. We show that as the distribution of data changes between the two domains, existing state-of-the-art fairness-preserving algorithms cannot match the performance they have on random data splits, where the source and target features are sampled from the same distribution. This observation demonstrates that model fairness is not naturally preserved under domain shift. Second, we present a novel algorithm that addresses domain shift when a fair outcome is of concern by combining fair model training via adversarial learning and and producing a shared domain-agnostic latent feature space for the source and target domains by minimizing the distance between the source and target embedding distributions. Through empirical evaluation, we show that combining our algorithms maintains fairness effectively under domain shift and also mitigates the effect of domain shift on the performance accuracy. Future extensions of this work includes considering scenarios where in addition to maintaining fairness under domain shift, the target domain maybe encountered sequentially, necessitating source-free model updating.

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

## A  Appendix

| Split | Source | | | | Target | | | |
|-------|------|-----|--------|--------|------|-----|--------|--------|
|       | Size | Y=0 | A=0\|Y=0 | A=0\|Y=1 | Size | Y=0 | A=0\|Y=0 | A=0\|Y=1 |
| A     | 34120 | 0.76 | 0.39 | 0.15 | 14722 | 0.76 | 0.39 | 0.15 |
| A1    | 12024 | 0.53 | 0.41 | 0.16 | 5393 | 0.91 | 0.49 | 0.18 |
| A2    | 29466 | 0.66 | 0.34 | 0.14 | 2219 | 0.97 | 0.48 | 0.30 |
| A3    | 11887 | 0.52 | 0.42 | 0.16 | 778 | 0.89 | 0.39 | 0.17 |
| C     | 3701 | 0.52 | 0.77 | 0.86 | 1577 | 0.54 | 0.76 | 0.84 |
| C1    | 2886 | 0.58 | 0.74 | 0.82 | 1096 | 0.67 | 0.78 | 0.86 |
| C2    | 903 | 0.47 | 0.80 | 0.80 | 96 | 0.74 | 0.70 | 0.92 |
| C3    | 2031 | 0.45 | 0.80 | 0.85 | 162 | 0.58 | 0.60 | 0.79 |
| G     | 697 | 0.70 | 0.28 | 0.37 | 303 | 0.70 | 0.30 | 0.34 |
| G1    | 573 | 0.66 | 0.34 | 0.45 | 427 | 0.76 | 0.23 | 0.20 |
| G2    | 388 | 0.61 | 0.36 | 0.49 | 196 | 0.84 | 0.20 | 0.16 |
| G3    | 439 | 0.62 | 0.35 | 0.45 | 159 | 0.87 | 0.21 | 0.19 |
| B     | 21184 | 0.87 | 0.02 | 0.06 | 9304 | 0.88 | 0.03 | 0.04 |
| B1    | 4259 | 0.87 | 0.00 | 0.01 | 8518 | 0.86 | 0.07 | 0.14 |
| B2    | 1314 | 0.90 | 0.00 | 0.01 | 5620 | 0.85 | 0.06 | 0.12 |
| B3    | 1560 | 0.88 | 0.01 | 0.04 | 10832 | 0.87 | 0.03 | 0.05 |

Table 8: Data split statistics. A,C,G,B correspond to the Adult, COMPAS, German and Bank dataset respectively. The rows with no number i.e. A,C,G,B correspond to random data splits. The numbered rows e.g. A1,A2,A3 correspond to statistics for specific splits. The columns represent the probabilities of specific outcomes for specific splits e.g. $P(Y = 0)$. Results when using *sex* as sensitive attribute, except for the Bank dataset, where *age* is the sensitive attribute.

### A.1  Data splits

The data splits employed in our approach are as follows:

**Adult Dataset**. We will use age, education and race to generate source and target domains. This can be a natural occurrence in practice, as gathered census information may differ along these axes geographically. For example, urban population is on average more educate than rural population [6], and more ethnically diverse [7]. Thus, a fair model trained on one of the two populations will need to overcome distribution shift when evaluated on the other population. Besides differences in the feature distributions, we also note the Adult dataset is both imbalanced in terms of outcome, $P(Y = 1) = 0.34$, and sensitive attribute of positive outcome, $P(A = 1|Y = 1) = 0.85$, i.e. only a fraction of participants are earning more than $50k$/year, and $85\%$ of them are male.

The source/target splits we consider are as follows:

1. Source data: White, More than 12 education years. Target data: Non-white, Less than 12 education years.

2. Source data: White, Older than 30. Target data: Non-white, younger than 40.

3. Source data: Younger than 70, More than 12 education years. Target data: Older than 70, less than 12 years of education.

In Table 8 we analyze the label and sensitive attribute conditional distributions for the above data splits. For the random split (A), the training and test label and conditional sensitive attribute distributions are identical, which is to be expected. For the three custom splits we observe all three distributions: $P(Y), P(A|Y = 0), P(A|Y = 1)$ differ between training and test. We also note the label distribution becomes more skewed towards $Y = 0$.

**COMPAS Dataset** Compared to the Adult dataset, the COMPAS dataset is balanced in terms of label distribution, however is imbalanced in terms of the conditional distribution of the sensitive attribute. We will split the dataset along age, number of priors, and charge degree, i.e. whether the person committed a felony or misdemeanor. Considered splits are as follows:

---

[6]https://www.ers.usda.gov/topics/rural-economy-population/employment-education/rural-education/
[7]https://www.ers.usda.gov/data-products/chart-gallery/gallery/chart-detail/?chartId=99538

1. Source data: Younger than 45, Less than 3 prior convictions. Test data: Older than 45, more than 3 prior convictions.

2. Source data: Younger than 45, White, At least one prior conviction. Target data: Older than 45, Non-white, No prior conviction.

3. Source data: Older than 25, At least one prior conviction, Convicted for a felony. Target data: Younger than 25, No priors, Convicted for a misdemeanor.

The first split tests whether a young population with limited number of convictions can be leveraged to fairly predict outcomes for an older population with more convictions. The second split introduces racial bias in the sampling process. In the third split we additionally consider the type of felony committed when splitting the dataset. For all splits, the test datasets become more imbalanced compared to the random split.

**German Credit Dataset** The dataset is smallest out of the three considered. For splitting we consider credit history and employment history. Similar to the Adult dataset, the label distribution is skewed towards increased risk i.e. $P(Y = 0) = 0.7$, and individuals of low risk are also skewed towards being part of the privileged group i.e. $P(A = 1|Y = 1) = 0.63$. We consider the following splits:

1. Source data: Employed up to 4 years. Target data: Employed long term (4+ years).

2. Source data: Up to date credit history, Employed less than 4 years. Target data: un-paid credit, Long term employed.

3. Source data: Delayed or paid credit, Employed up to 4 years. Target data: Critical account condition, Long term employment.

Compared to random data splits, the custom splits reduce label and sensitive attribute imbalance in the source domain, and increase these in the target domain.

**Bank Marketing Dataset** The dataset records the effects of a marketing campaigns initiated by a bank on its term deposits. Compared to the other datasets, the bank dataset is highly imbalanced, both in terms of label distribution and sensitive attribute distribution. We consider the following splits:

1. Source data: Made a loan, has a job. Target data: No loan, unemployed.

2. Source data: Married, not self employed. Target data: Not married, self employed.

3. Source data: Followed a professional course, Married, Technician. Target data: High School educated, Not Married, Blue-collar job.

## A.2  Parameter tuning and implementation

### A.2.1  Training and model selection

Implementation of our approach is done using the PyTorch Paszke et al. (2019) deep learning library. We model our encoder $e_u$ as a one layer neural network with output space $z \in \mathbb{R}^{20}$. Classifiers $g$ and $h$ are also one layer networks with output space $\in \mathbb{R}^2$. We train our model for $45,000$ iterations, where the first $30,000$ iterations only involve source training. For the first $15,000$ we only perform minimization of the binary cross entropy loss $\mathcal{L}_{bce}$. We introduce source fairness training at iteration $15,000$, and train the fair model, i.e. with respect to both $\mathcal{L}_{bce}$ and $\mathcal{L}_{fair}$, for $15,000$ more iterations. In the last $15,000$ iterations we perform adaptation, where we optimize $\mathcal{L}_{bce}, \mathcal{L}_{fair}$ on the source domain, $\mathcal{L}_{fair}$ on the target domain, and $\mathcal{L}_{swd}$ between the source and target embeddings $e_u((x^s, a^s)), e_u((x^t, a^t))$ respectively. We use a learning rate for $\mathcal{L}_{bce}, \mathcal{L}_{fair}$ of $1e - 4$, and learning rate for $\mathcal{L}_{swd}$ of $1e - 5$. Model selection is done by considering the difference between accuracy on the validation set, and demographic parity on the test set. Given equalized odds and averaged opportunity require access to the underlying labels on the test set we cannot use these metrics for model selection. Additionally, models corresponding to degenerate predictions i.e. test set predicted labels being either all 0s or all 1s are not included in result reporting.

### A.3 Empirical Results about Dynamics of Learning

We performed another analytic experiment to study the effect of model training on the important loss terms and metric. In Figure 3, we analyze the effect of the adaptation process on target domain accuracy, validation accuracy, demographic parity on the source domain, and demographic parity on the target domain for the Adult dataset. We compare two scenarios: (1) running the algorithm when $\mathcal{L}_{swd}$ is not enforced (bottom), and (2) running the algorithm using both fairness and domain alignment (top). For the first $30,000$ iterations, we only perform source-training, where the first half of iterations is spent optimizing $\mathcal{L}_{bce}$, and the second half is spent jointly optimizing $\mathcal{L}_{bce}$ and the source fairness objective. We note once optimization with respect to $\mathcal{L}_{fair}$ starts, demographic parity decreases until adaptation start, i.e., iterations $15,000$ to $30,000$. The validation accuracy in this interval also slightly decreases, as improving fairness may affect accuracy performance. During adaptation, i.e., after iteration $30,000$, we observe that in the scenario where we use $\mathcal{L}_{swd}$, the target domain accuracy increases, while demographic parity on both the source and target domains remains relatively unchanged. In the scenario where no optimization of $\mathcal{L}_{swd}$ is performed, there is still improvement with respect to target accuracy. However, target domain demographic parity becomes on average larger. These observations imply that the distributional alignment at the output of the encoder has beneficial effects both for the classification as well as the fairness objective and our algorithm gradually leads to the desired effects.

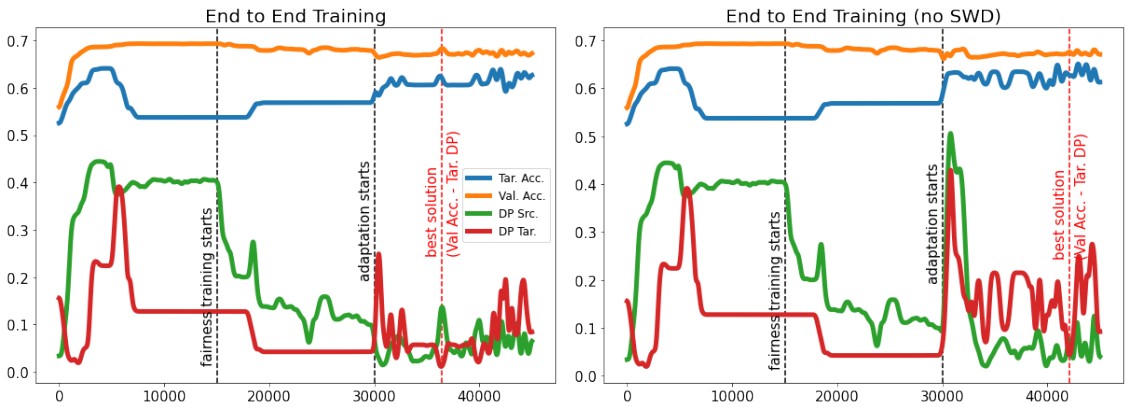

Figure 3: Learning behavior during training when using both $\mathcal{L}_{fair}$ and $\mathcal{L}_{swd}$ (left) versus when only using $\mathcal{L}_{fair}$ (right)

We further investigate the different components present in our algorithm. In Figure 3 we analyze the training and adaptation process with respect to target accuracy, validation accuracy, demographic parity on the source domain, and demographic parity on the target domain. Performance plots are reported for the Adult dataset. We compare two scenarios: running the algorithm when $\mathcal{L}_{swd}$ is not enforced (bottom), and running the algorithm using both fairness and domain alignment (top). For the first $30,000$ iterations we only perform source training, where the first half of iterations is spent optimizing $\mathcal{L}_{bce}$, and the second half is spent jointly optimizing $\mathcal{L}_{bce}$ and the source fairness objective. We note once optimization with respect to $\mathcal{L}_{fair}$ starts, demographic parity decreases until adaptation start, i.e. between iterations $15,000 - 30,000$. The validation accuracy in this interval also slightly decreases, as improving fairness may affect accuracy performance. During adaptation, i.e. after iteration $30,000$, we observe that in the scenario where we use $\mathcal{L}_{swd}$, the target accuracy increases, while demographic parity on both source and target domains remains relatively unchanged. In the scenario where no optimization of $\mathcal{L}_{swd}$ is performed, there is still improvement with respect to target accuracy, however target demographic parity becomes on average larger. This implies that the distributional alignment loss done at the output of the encoder has beneficial effects both for the classification as well as the fairness objective.

### A.4 Additional dataset analysis

Similar to the analysis in the main body of the paper, we evaluate performance on the Bank dataset and report results in Table 9.

On all data splits our approach leads to the best outcome in terms of $\Delta DP$, and on the first two splits we also achieve highest accuracy amongst the fairness preserving methods. Moreover, besides our approach, only RP is able

to strike a balance between fairness and accuracy on all splits, and our approach proves superior in terms of accuracy and demographic parity. We also note that compared to the other datasets, $\Delta EO$ and $\Delta AO$ are not automatically improved with the optimization of $\Delta DP$. This is the case with all other methods as well - either competitive accuracy or several fairness metrics will not be enforced. For our method, the sensitivity of $\Delta EO$ and $\Delta AO$ appears to be high, while that of the accuracy is low. This suggests that these metrics may be further improved with higher focus on dataset specific hyper-parameter tuning.

| Alg. | Admin., Married | | | | Technician, Married, Housing | | | | Technician, Education, Housing | | | |
|---|---|---|---|---|---|---|---|---|---|---|---|---|
| | Acc. | $\Delta DP$ | $\Delta EO$ | $\Delta AO$ | Acc. | $\Delta DP$ | $\Delta EO$ | $\Delta AO$ | Acc. | $\Delta DP$ | $\Delta EO$ | $\Delta AO$ |
| Base | $0.82^{\pm 0.01}$ | $0.00^{\pm 0.01}$ | $0.21^{\pm 0.07}$ | $0.17^{\pm 0.01}$ | $0.83^{\pm 0.05}$ | $0.03^{\pm 0.05}$ | $0.07^{\pm 0.12}$ | $0.06^{\pm 0.06}$ | $0.83^{\pm 0.02}$ | $0.12^{\pm 0.03}$ | $0.54^{\pm 0.05}$ | $0.30^{\pm 0.03}$ |
| MC | $0.79^{\pm 0.00}$ | $0.06^{\pm 0.07}$ | $0.15^{\pm 0.11}$ | $0.09^{\pm 0.05}$ | $0.72^{\pm 0.02}$ | $0.09^{\pm 0.04}$ | $0.12^{\pm 0.06}$ | $0.12^{\pm 0.03}$ | $0.88^{\pm 0.01}$ | $0.01^{\pm 0.01}$ | $0.37^{\pm 0.02}$ | $0.18^{\pm 0.02}$ |
| AD | $0.58^{\pm 0.08}$ | $0.28^{\pm 0.25}$ | $0.52^{\pm 0.31}$ | $0.34^{\pm 0.26}$ | $0.56^{\pm 0.02}$ | $0.30^{\pm 0.38}$ | $0.41^{\pm 0.36}$ | $0.33^{\pm 0.37}$ | $0.54^{\pm 0.04}$ | $0.12^{\pm 0.12}$ | $0.23^{\pm 0.14}$ | $0.16^{\pm 0.11}$ |
| ROC | $0.51^{\pm 0.00}$ | $0.06^{\pm 0.00}$ | $0.00^{\pm 0.00}$ | $0.05^{\pm 0.00}$ | $0.81^{\pm 0.00}$ | $0.22^{\pm 0.00}$ | $0.06^{\pm 0.00}$ | $0.12^{\pm 0.00}$ | $0.61^{\pm 0.00}$ | $0.06^{\pm 0.00}$ | $0.02^{\pm 0.00}$ | $0.03^{\pm 0.00}$ |
| EGR | $0.70^{\pm 0.01}$ | $0.06^{\pm 0.03}$ | $0.28^{\pm 0.11}$ | $0.17^{\pm 0.06}$ | $0.61^{\pm 0.01}$ | $0.06^{\pm 0.03}$ | $0.25^{\pm 0.01}$ | $0.12^{\pm 0.03}$ | $0.68^{\pm 0.02}$ | $0.02^{\pm 0.01}$ | $0.16^{\pm 0.04}$ | $0.08^{\pm 0.03}$ |
| LFR | $0.68^{\pm 0.05}$ | $0.04^{\pm 0.04}$ | $0.10^{\pm 0.05}$ | $0.08^{\pm 0.05}$ | $0.63^{\pm 0.07}$ | $0.06^{\pm 0.04}$ | $0.21^{\pm 0.16}$ | $0.09^{\pm 0.08}$ | $0.67^{\pm 0.07}$ | $0.07^{\pm 0.06}$ | $0.32^{\pm 0.23}$ | $0.17^{\pm 0.12}$ |
| CEO | $0.61^{\pm 0.10}$ | $0.01^{\pm 0.01}$ | $0.03^{\pm 0.01}$ | $0.03^{\pm 0.02}$ | $0.50^{\pm 0.00}$ | $0.00^{\pm 0.00}$ | $0.00^{\pm 0.00}$ | $0.00^{\pm 0.00}$ | $0.60^{\pm 0.05}$ | $0.05^{\pm 0.04}$ | $0.11^{\pm 0.08}$ | $0.07^{\pm 0.04}$ |
| RP | $0.75^{\pm 0.00}$ | $0.00^{\pm 0.00}$ | $0.10^{\pm 0.00}$ | $0.09^{\pm 0.00}$ | $0.80^{\pm 0.00}$ | $0.02^{\pm 0.00}$ | $0.21^{\pm 0.00}$ | $0.06^{\pm 0.00}$ | $0.84^{\pm 0.00}$ | $0.05^{\pm 0.00}$ | $0.33^{\pm 0.00}$ | $0.19^{\pm 0.00}$ |
| Ours | $0.81^{\pm 0.01}$ | $0.00^{\pm 0.00}$ | $0.25^{\pm 0.10}$ | $0.17^{\pm 0.02}$ | $0.84^{\pm 0.01}$ | $0.00^{\pm 0.00}$ | $0.26^{\pm 0.29}$ | $0.11^{\pm 0.12}$ | $0.84^{\pm 0.02}$ | $0.01^{\pm 0.01}$ | $0.55^{\pm 0.11}$ | $0.23^{\pm 0.04}$ |

Table 9: Performance results for the three splits of the bank dataset

