# OpenReview forum: "Preserving Fairness in AI under Domain Shift"
_TMLR — Rejected by TMLR_

### Review · Reviewer_L7d8 · 2023-08-23

**Summary Of Contributions:**

The manuscript demonstrates that existing approaches to enforcing fairness constraints fail in some cases where there is distribution shift between the training data and test data. In addition, it is shown how a standard unsupervised domain adaptation concept (minimising some notion of distribution distance) can be combined with a fairness objective to ameliorate the issue.

**Audience:**

Yes

**Broader Impact Concerns:**

There is no explicit inclusion of a broader impact section, but there is some discussion of this due to the nature of the research. It could be useful to have a section dedicated to this.

**Claims And Evidence:**

No

**Requested Changes:**

The (statistical) significance/analysis of the work should be improved. Averaging the performance measurements over some number of runs is insufficient. Some notion of uncertainty of the numbers should also be incorporated, and the source of uncertainty should be identified.

**Strengths And Weaknesses:**

* I found the manuscript well-written and easy to follow. In particular, I think the paper does a good job of reviewing previous work from several different relevant communities.
* The problem studied in this work is important in practice, and following in the footsteps of UDA seems like a good first step. In many cases distribution shift can happen gradually---could the authors comment on the applicability, or limitations, of their proposed method in such a context?
* The proposed method is simple and based on well-understood UDA and fairness ideas.
* The experimental evaluation could be improved in a few ways: (i) there are no error bars, so it is unclear how reliable the results are; (ii) the authors state that they average over 7 runs, but do not mention what is different across each run, so it is not clear what type of uncertainty is being accounted for; and (iii) the distribution shift between the source and target domains is often quite extreme---e.g., disjoint races and sexes. I think it would be more realistic to consider smaller distribution shifts, e.g., on the adult dataset by including white and non-white in both source and target domains, but at different frequencies.

---

> ### Author Response · Authors · 2023-09-16
> **Author reply**
>
> We thank the reviewer for their recommendation. We are glad that the reviewer have found our problem setting important and our experiments convincing. We have updated all the result tables in our paper and included a standard error measurement. We would like to stress that previous values also were average values and we only rerun our experiments and added the standard deviation values. We have discussed the implications result from this in the main body of the paper under results.

---

### Review · Reviewer_EEEE · 2023-08-29

**Summary Of Contributions:**

The paper studies the transferability fairness under domain shifts. It considers a setting of domain adaptation when there exists distribution shifts between training and testing data. The goal is to train a model from source domain that is fair and accurate on target domain. The paper proposes an algorithm that learns the domain-invariant unbiased representations. The experiments are also conducted to validate the proposed method.

**Audience:**

Yes

**Claims And Evidence:**

Yes

**Requested Changes:**

1. Add more extensive literature review and discuss the differences from the existing methods.
2. Explain what makes the proposed algorithm novel compare to the existing methods.
3. Add more empirical studies to compare with more baselines on more datasets

**Strengths And Weaknesses:**

1. Existing works are not sufficiently discussed and their differences are unclear: There are already many works in the literature studying the transferability/generalization of fairness in ML. Although the paper has mentioned a few in the related work section, many are not yet mentioned and the authors should discuss the differences with these works in detail, e.g.,
    [1]. Taeho Yoon, Jaewook Lee, and Woojin Lee. Joint transfer of model knowledge and fairness over domains using wasserstein distance. IEEE Access, 8:123783–123798, 2020.
    [2]. Candice Schumann, Xuezhi Wang, Alex Beutel, Jilin Chen, Hai Qian, and Ed H Chi. Transfer of machine learning fairness across domains. arXiv preprint arXiv:1906.09688, 2019
    [3]. David Madras, Elliot Creager, Toniann Pitassi, and Richard Zemel. Learning adversarially fair and transferable representations. In International Conference on Machine Learning, pp. 3384–3393. PMLR, 2018.
    [4]. Luca Oneto, Michele Donini, Andreas Maurer, and Massimiliano Pontil. Learning fair and transferable representations. arXiv preprint arXiv:1906.10673, 2019.
    [5]. Amanda Coston, Karthikeyan Natesan Ramamurthy, Dennis Wei, Kush R Varshney, Skyler Speakman, Zairah Mustahsan, and Supriyo Chakraborty. Fair transfer learning with missing protected attributes. In Proceedings of the 2019 AAAI/ACM Conference on AI, Ethics, and Society, pp. 91–98, 2019.
    [6]. Pham, Thai-Hoang, Xueru Zhang, and Ping Zhang. "Fairness and Accuracy under Domain Generalization." In The Eleventh International Conference on Learning Representations. 2022.
2. Novelty of the proposed method is not significant: To ensure fairness and accuracy attained during training to be transferred to the target domain, the proposed leverage adversarial learning and domain-invariant representation learning, where adversarial learning is used to learn fair representations independent of sensitive attribute while the invariant representation learning ensures the transferability of fairness by aligning source and target domain. As far as I know, both techniques are standard in fair ML and domain adaptation literature.
3. The proposed algorithm is only evaluated empirically while there is no theoretical guarantee. Moreover, the algorithm is only compared with a few works on three datasets. It is critical to compare the algorithm with more recent works mentioned above.

---

> ### Author Response · Authors · 2023-09-16
> **Author Reply**
>
> We thank the reviewer for the comments that helped us to improve our manuscript as we tried to include all the requested changes in the manuscript. We hope through continual discussion, we can address the concerns raised by the reviewer.
>
> We have updated the related work section between lines to include the works referenced by the reviewer. Please note the References [2],[3], [5] that the reviewer mentioned were already included in our related work discussion, and [6] is not a peer reviewed work. While these works handle the problem of transfer learning under fairness restrictions, the application and problem setup differ from our work. We respectfully ask the reviewer to compare these works side-by-side with our work to ensure that the differences between our work and these works are significant:
>   - In [1], the authors consider a scenario where a fair classifier trained on a source domain is deployed on a target domain where the sensitive attribute changes. We differ from this work through our application setting, experimental design and our choice of ensuring fairness - using adversarial learning. [1] consider random data splits for source and target domains with different sensitive attributes. This leads to the same data distribution between the source and target domain, and only the sensitive attribute distribution changes. We maintain the same sensitive attribute, however engineered data splits to manifest a large distributional shift between the source and target domains.
>   - In [2], a semi-supervised learning setting is considered, with the goal of ensuring fairness under a change in the sensitive attribute between source and target domains. This again differs from our approach, as we consider a fully unsupervised target domain. Please note this work was already included in our related work section.
>   - [3] propose an algorithm that which learns fair representations. However, the proposed approach does not consider performance under distributional shift. Please note this work was already included in our related work section.
>   - [4] propose to improve model fairness and generalization to new domains by considering a multi-task learning framework. The tasks are however randomly generated from the available data, limiting the distributional drift the model has to overcome.
>   - [5] propose a fairness preserving approach where the sensitive attributes are not observed. This differs from the problem setting considered in our work and the works we compare against, where the main difficulty stems from the presence of domain shift between source and target distributions. Please not this work was already included in our related work section.
>   - [6] propose a multi source fairness preserving approach, where an algorithm leverages several source domains in order to ensure fairness and generalization on a target domain. However, compared to our work, data splits considered are based loosely on separating the data into age groups or image rotations, which may not significantly produce differences between the conditional probability distributions between labels and sensitive attributes across source and target domains.
>
> Compared to the approaches mentioned above, and the other works we refer to in the main body of the paper, our algorithm has been proven to handle distributional shift while maintaining fairness on data splits which have been engineered to specifically increase the difficulty of the adaptation problem. Besides empirical performance, we additionally demonstrate that combining the minimization of Sliced Wasserstein Distance for minimizing distributional shift, with adversarial training for fairness leads to a viable algorithm. We mention that while these techniques have been independently used in the past, it is non trivial to demonstrate their performance for our considered application domain.
>
> We have followed the reviewers' suggestion, and added experiments on another dataset, the Bank Marketing dataset, and included three more baselines in all our results. The previously observed conclusions on the superiority of our method are maintained. We also attempted to compare against some of the methods proposed by the reviewer, but only found available code for [3] and [6]. The code for [6] tackles image datasets, and is not suited for tabular classification tasks. We were able to run the code in [3] on the splits in the adult dataset, however the results we obtained showed the model produce degenerate results on all three splits (.5 balanced accuracy, 0 demographic parity). This suggests that other approaches may not be ready to be deployed out of the box on our application domain, and may require additional fine tuning.

---

### Review · Reviewer_xR7D · 2023-08-30

**Summary Of Contributions:**

This submission proposes an algorithm to preserve model fairness under distribution shift. Previous fairness-preserving methods mainly consider the standard setting whether test and training distribution are almost the same or have small differences. However, the source and target domains may differ, and labels only exist in the source domain. Purely applying domain adaptation methods on the target domain enlarges the model bias. This submission solves this setting, by combining the unsupervised domain adaptation (to improve generalization) with adversarial training based distribution alignment (to improve fairness). Experiments on Adult, German, and COMPAS show the effectiveness.

**Audience:**

Yes

**Broader Impact Concerns:**

Authors may need to discuss broader impacts, e.g., the scope, implications, and limitations of current fairness definitions.

**Claims And Evidence:**

No

**Requested Changes:**

1. It would be great if the writing quality could be improved and several grammatical errors and unclear points could be explained.
2. The detail discussion on hyperparameter selection guidelines is appreciated, along with training efficiency discussions.

**Strengths And Weaknesses:**

Strengths:
- The submission studies an important problem in practice, where existing fairness-preserving methods mainly consider aligned training and test distributions and this submission addresses the fairness-preserving problem when there is a domain shift.
- The proposed algorithm works effectively in practice.

Weaknesses:
- The proposed algorithm is relatively straightforward --- the adversarial training scheme mainly requires sensitive attributes for preserving fairness, i.e., without supervised labels. So it can be directly applied in domain adaptation settings. At the same time, the standard unsupervised domain adaptation regime can be applied as well. Combining both techniques yields the proposed algorithm as shown in Eqn. (6). Moreover, such a combination results in several hyperparameters to tune, leading to nontrivial burdens on the user side.

- The writing quality can be further improved. There are several typos and grammatical errors (as listed below). Some experimental results do not match text illustrations.

Typos and grammatical errors:
1. Line 55: "a persistent tasks"
2. Line 57: "perofrmnace"
3. Line 203: "a fair mode"
4. Line 224: "tat"
5. Confusing item number in Line 237 and Line 239.
6. Line 329: "trad-off"
7. Line 398 "Finally, SWD achieves the best perofrmnace in the best perofrmnace but at 398 the cost of fairness."
8. Line 604, Figure 3: there is no "top" and "bottom" subfigure.

Unclear points:
1. Is Eqn. (4) the loss on one data point or the whole distribution?
2. From Table 1, it is not clear "common UDA methods would fail to preserve fairness". It is the ground-truth data distribution shown.
3. Line 337-338, "our method leads to the 337 best accuracy performance compared to the methods that maintain fairness". In Table 2, "Ours" line does not consistently achieve the best accuracy.
4. Line 350-351, "our method is able to maintain fairness after adaptation while being competitive in terms of accuracy performance". In Table 3, "Age, Education" column, "ours" has significant degradation on accuracy, which does not support "being competitive in terms of accuracy performance".
5. Does the baseline method consider domain shift? It seems not, but if this is the case, our method should have better accuracy, which contradicts the results in Table 5.
6. Line 395 "On the second split we observe the SWD only model has 395 poorest performance," contradicts Line 398 "Finally, SWD achieves the best perofrmnace in the best perofrmnace but at 398 the cost of fairness."

Minor:
1. The first paragraph in Section 1 is a bit long and away from the topic. Maybe it's better to introduce the fairness issue more directly.

---

> ### Author Response · Authors · 2023-09-16
> **Author reply**
>
> We thank the reviewer for the careful reading of our work. We are glad that the review is recognizing the importance of the setting we study and the effectiveness of the proposed solution.
>
> In response to your concern about presentation, we have corrected the grammatical errors. Point 5: We wish to denote 4,5 (lines 237-239) as an addition to points 1,2,3 (lines 204-211). We have noted this under lines 235-236.
>
> For the Unclear Points:
> - The reviewer is correct, the SWD is a distributional distance metric, as explained in lines 214-229. It is computed   over the whole distributions but the computation is point-wise based on samples from the two distributions. We have referred the reader for these details to Redko et al. (2017) but we can add further details per the reviewer's advice.
> - Common UDA methods rely on establishing a shared embedding space for both the source and target distributions. These approaches typically prioritize domain-invariance and are agnostic to sensitive attribute conditional probabilities necessary for maintaining prediction fairness. We have updated the text to reflect this point.
> - We have updated the result tables to add standard errors, and the text to  reflect performance.
> - We have updated the results and the text.
> - It is expected the baseline method outperforms our method in terms of accuracy only on random data splits, as presented in Table 2. The reason is that the baseline only considers accuracy in its optimization while the optimization loss in our method involves fairness metrics as well. Given the other tables explore performance under domain shift, it may be the case that the baseline methods will observe deteriorated accuracy performance.
> - We have updated the results and the text.
> - We have rephrased paragraph 1 to be more succinct and introduce the fairness problem.

---

> > ### Comment · Reviewer_xR7D · 2023-09-20
> >
> > Thanks for the response and most of my concerns are addressed.
> >
> > However, it still seems strange to me that in Tables 3,4,5, the baseline method sometimes perform better in terms of accuracy compared to the proposed method, even though the baseline method does not consider the domain shifting.
> >
> > Moreover, the overhead of tuning several hyperparameters seems to be intrinsic for this method.

---

> > > ### Author Response · Authors · 2023-09-21
> > > **Author reply**
> > >
> > > Dear reviewer,
> > >
> > > Thank you for your continued engagement with our work.
> > >
> > > As observed, in Tables 3,4,5 the baseline method observes higher accuracy than our approach. Usually, when improving the fairness of a model, there exists a tradeoff between fairness and accuracy. This phenomenon has been studied in the literature. We would like to respectfully ask you to consider checking the following references [1], [2], [3], and [4], where you can observe a similar trend. Hence, our results are not exceptions and follow well-established observations. As you also expect, we would like to see no accuracy decline compared to the baseline approach and just an improvement in fairness in an ideal situation, but in most cases, a trade-off will be observed because fairness and accuracy losses can potentially work against each other. Note this is not a rule, as Table 5 has examples of both improved accuracy and fairness. This observation illustrates that our algorithm has the potential of detecting a more robust and fair classifier, however, the guarantee that we offer is that an unfair classifier becomes fairer while maintaining performance on a differently distributed target domain.
> > >
> > > In regards to hyperparameter tuning, we are required to fix the regularizer for the fairness and distributional distance losses, coupled with the total number of training iterations. Note in our experiments we have used a fairness loss regularizer of 1 for all experiments, so our algorithm seems robust in regards to this.
> > >
> > >
> > > [1] - Wang et. al., KDD '21 - https://dl.acm.org/doi/abs/10.1145/3447548.3467326
> > > [2] - Li et. al., AAAI '23 - https://ojs.aaai.org/index.php/AAAI/article/view/26674
> > > [3] - Wang et. al., ICML '22 - https://proceedings.mlr.press/v162/wang22ac.html
> > > [4] - Madras et. al., ICML '18 - https://proceedings.mlr.press/v80/madras18a/madras18a.pdf

---

### Decision · Action_Editors · 2023-10-05

**Recommendation:** Reject

**Comment:**

This paper has mixed opinions from the reviewers, while the negative opinions outweigh the positive opinions. I carefully read the entire reviews, corresponding responses, and the revised paper. As my comment in "Claims And Evidence", I think this paper has a large room for improvement in validating the proposed method's effectiveness. One possible option is to add theoretical grounding of the proposed method (e.g., why fairness methods will fail in UDA settings and why UDA methods will fail in algorithmic fairness settings). This approach will need a rigorous definition of "fairness" (e.g., which fairness metric is targeted by the proposed method?) and the significance of the distribution shifts (e.g., "domain discrepancy" defined as a divergence between two domain distributions [A]). Another possible option is to show empirical outperforming performance compared to a number of the existing recent fairness and UDA methods on the target benchmark. The current comparisons are mainly done by old methods (before 2019), while there have been a number of fairness and UDA works since 2019 (I do not list them all here).

[A] Ben-David, Shai, et al. "A theory of learning from different domains." Machine learning 79 (2010): 151-175.

Overall, I am persuaded by the negative opinions rather than the positive ones. My final recommendation is to reject this submission in its current form.

**Audience:**

This paper bridges the gap between unsupervised domain adaptation (UDA) and algorithmic fairness. This paper is relevant to those working in algorithmic fairness and UDA, but as the reviewers pointed out, this paper missed somewhat relevant papers in the area.

**Claims And Evidence:**

This paper addresses the fairness problem under domain shift based on Unsupervised Domain Adaptation (UDA). The proposed algorithm is based on the minimax optimization for updating the target classifier, and feature extractor encoder and the sensitive attribute classifier. The experimental results are shown in tabular datasets, such as Adult, German, COMPAS and Bank Marketing.

As Reviewer EEEE's comment, this paper only shows empirical evaluations not showing theoretical guarantee, and only compared with a few works on four datasets. I generally agree with Reviewer EEEE's comment; it is not the issue of the number of datasets (they are common fairness benchmarks), but I think it could contain more evaluations with other comparison methods and it may needs additional analysis on the method rather than showing UMAP visualization (e.g., the impact of the hyperparameters as Reviewer xR7D's comment).

I think this paper could be improved by additional analyses (including theoretical guarantee, comparisons with more methods, hyperparameter study, in-depth analyses of the proposed modules, etc.).